# Neuropathic MORC2 mutations perturb GHKL ATPase dimerization dynamics and epigenetic silencing by multiple structural mechanisms

Christopher H. Douse [1], Stuart Bloor[2], Yangci Liu[1], Maria Shamin[1], Iva A. Tchasovnikarova [2,3], Richard T. Timms[2,4], Paul J. Lehner[2] & Yorgo Modis [1]

Missense mutations in *MORC2* cause neuropathies including spinal muscular atrophy and Charcot–Marie–Tooth disease. We recently identified MORC2 as an effector of epigenetic silencing by the human silencing hub (HUSH). Here we report the biochemical and cellular activities of MORC2 variants, alongside crystal structures of wild-type and neuropathic forms of a human MORC2 fragment comprising the GHKL-type ATPase module and CW-type zinc finger. This fragment dimerizes upon binding ATP and contains a hinged, functionally critical coiled-coil insertion absent in other GHKL ATPases. We find that dimerization and DNA binding of the MORC2 ATPase module transduce HUSH-dependent silencing. Disease mutations change the dynamics of dimerization by distinct structural mechanisms: destabilizing the ATPase-CW module, trapping the ATP lid, or perturbing the dimer interface. These defects lead to the modulation of HUSH function, thus providing a molecular basis for understanding MORC2-associated neuropathies.

---

[1] Department of Medicine, MRC Laboratory of Molecular Biology, Cambridge Biomedical Campus, University of Cambridge, Cambridge, CB2 0QH, UK. [2] Department of Medicine, Cambridge Institute for Medical Research, Cambridge Biomedical Campus, University of Cambridge, Cambridge, CB2 0XY, UK. [3] Department of Molecular Biology, Massachusetts General Hospital, and Department of Genetics, Harvard Medical School, Boston, MA 02114, USA. [4] Department of Medicine, Brigham and Women's Hospital, Boston, MA 02115, USA. Correspondence and requests for materials should be addressed to C.H.D. (email: cdouse@mrc-lmb.cam.ac.uk) or to Y.M. (email: ymodis@mrc-lmb.cam.ac.uk)

Microrchidia CW-type zinc finger proteins (MORCs) are a family of transcriptional regulators conserved in eukaryotes. More specifically, MORCs regulate the epigenetic control of transposons and newly integrated transgenes at different developmental stages in plants[1,2], nematodes[1,3], and mammals[4,5]. Four mammalian genes (MORC1–4) have been annotated. MORC1 is required for spermatogenesis in mice[6], as an effector of transposon silencing[5]. We recently showed that human MORC2 is necessary, in conjunction with the human silencing hub (HUSH), for silencing of transgenes integrated at chromatin loci with histone H3 trimethylated at lysine 9 H3K9me3[4,7]. HUSH and MORC2 were further found to restrict transposable elements from the long interspersed element-1 class[8]. MORC2 has also been reported to have ATP-dependent chromatin remodeling activity, which contributes to the DNA damage response[9] and to downregulation of oncogenic carbonic anhydrase IX in a mechanism dependent on histone deacetylation by HDAC4[10]. MORC3 localizes to H3K4me3-marked chromatin, but the biological function of MORC3 remains unknown[11].

Despite growing evidence of their importance as chromatin regulators, MORCs have been sparsely characterized at the molecular level. Mammalian MORCs are large, multidomain proteins, with an N-terminal gyrase, heat shock protein 90, histidine kinase and MutL (GHKL)-type ATPase module, a central CW-type zinc finger (CW) domain, and a divergent C-terminal region with one or more coiled coils that are thought to enable constitutive dimerization[12]. Structural maintenance of chromosomes flexible hinge domain-containing protein 1 (SMCHD1) shares some of these key features and could therefore be considered as a fifth mammalian MORC, but it lacks a CW domain, and has a long central linker connecting to an SMC-like hinge domain[13]. As with several other members of the GHKL superfamily, the ATPase module of MORC3 dimerizes in an ATP-dependent manner[11]. The recently reported crystal structure of the ATPase-CW cassette from mouse MORC3 consists of a homodimer, with the non-hydrolysable ATP analog AMPPNP and an H3K4me3 peptide fragment bound to each protomer[11]. The trimethyl-lysine of the H3K4me3 peptide binds to an aromatic cage in the CW domains of MORC3 and MORC4[11,14,15]. The MORC3 ATPase domain was also shown to bind DNA, and the CW domain of MORC3 was proposed to autoinhibit DNA binding and ATP hydrolysis by the ATPase module[15]. Based on the observed biochemical activities, MORCs have been proposed to function as ATP-dependent molecular clamps around DNA[11]. However, the CW domains of MORC1 and MORC2 lack the aromatic cage and do not bind H3K4me3, suggesting that different MORCs engage with chromatin via different mechanisms[4,14]. Moreover, MORC1 and MORC2 contain additional domains, including a predicted coiled-coil insertion within the ATPase module that has not been found in any other GHKL ATPases.

Exome sequencing data from patients with genetically unsolved neuropathies have recently reported missense mutations in the ATPase module of the MORC2 gene[16–23]. A range of symptoms have been detailed, all subject to autosomal dominant inheritance, with a complex genotype–phenotype correlation. Several reports describe Charcot–Marie–Tooth (CMT) disease in families carrying MORC2 mutations including R252W (most commonly)[16,17,20,21]; patients presented in the first or second decade with distal weakness that spread proximally, usually accompanied by signs of CNS involvement. Two other mutations, S87L and T424R, have been reported to cause congenital or infantile onset of neuropathies[16,19,21,22]. Severe spinal muscular atrophy (SMA) with primary involvement of proximal muscles and progressive cerebellar atrophy was detailed in patients with the T424R mutation[19,22], while diagnosis of patients with the S87L mutation

was CMT with SMA-like features[16,21]. We recently showed that the CMT-associated MORC2 mutation R252W hyperactivates HUSH-mediated epigenetic silencing in neuronal cells[4]. Disease mutations in MORC2 map to the ATPase module, as in the related SMCHD1 protein, where mutations have recently been associated with Bosma arhinia microphthalmia syndrome (BAMS)[24,25]. However, the lack of biochemical or structural data on MORC2 has precluded the efforts to understand its molecular function and rationalize the phenotypes of this rapidly growing list of neuropathic MORC2 variants.

Here we report the biochemical and cellular activities of MORC2 variants, alongside structures of wild-type and neuropathic variant forms of a MORC2 fragment comprising the GHKL-type ATPase module and CW domain. Our data reveal several functionally critical features of MORC2. We show how the ATPase activity of MORC2 is tuned, and how ATP-dependent dimerization and DNA binding transduce HUSH-dependent silencing. We propose distinct structural mechanisms that explain how these functions of MORC2 are misregulated in neuropathy-associated variants: destabilization of the ATPase-CW module, trapping the ATP lid, or perturbing the dimer interface. Together, our data provide a molecular understanding of the multiple structural mechanisms underlying the neuropathic effects of MORC2 mutations.

## Results

**Isolation of an active GHKL ATPase module of MORC2.** We set out to purify human MORC2 constructs suitable for biochemical and structural studies. MORC2 is a 1032-amino acid protein predicted to contain several functional domains including an N-terminal GHKL-type ATPase module with a coiled-coil insertion (CC1) (Fig. 1a). MORC3 lacks the CC1 insertion, and its ATPase module can be produced in E. coli[11,15], but we were unable to purify soluble human MORC2 with an intact CC1 from bacteria. We purified MORC2(1–282) (i.e., the GHKL domain, but not including CC1 or the remainder of the ATPase module) and used native differential scanning fluorimetry (DSF) to monitor its interaction with non-hydrolysable ATP analog AMPPNP. Native DSF exploits the intrinsic fluorescence properties of proteins to monitor thermal unfolding processes, which are usually accompanied by a redshift in the fluorescence maximum wavelength from 330 nm to 350 nm. By monitoring the ratio of fluorescence at these wavelengths as a function of temperature, a melting temperature ($T_m$) may be extracted in the absence and presence of stabilizing ligands. We found that although we could detect a small thermal stabilization of MORC2 (1–282) by $Mg^{2+}$/AMPPNP, indicating an interaction, no ATPase activity was detected using this construct in an endpoint assay based on the detection of inorganic phosphate (Fig. 1b and Supplementary Fig. 1a). We could express and purify full-length MORC2 bearing a cleavable tandem StrepII tag in insect cells using a baculovirus-based expression system, but obtained low yields, and the protein was highly sensitive to proteolytic cleavage. We used limited proteolysis to identify a protease-resistant N-terminal fragment that ran at ~75 kDa by SDS-PAGE and showed evidence of ATP hydrolysis activity (Supplementary Fig. 1a,b). Based on the sequencing of tryptic peptides by tandem mass spectrometry, we designed a construct spanning residues 1–603, encompassing the N-terminal ATPase module, the CW domain, and the second predicted coiled coil (CC2).

MORC2(1–603) from insect cells was stabilized by $Mg^{2+}$/AMPPNP, leading to a large increase in the protein $T_m$ from 51.5 to 67.3 °C as measured by DSF (Fig. 1b and Supplementary Fig. 1c). By contrast, concentrations up to 2 mM of $Mg^{2+}$/ADP and inorganic phosphate (the products of ATP hydrolysis) did

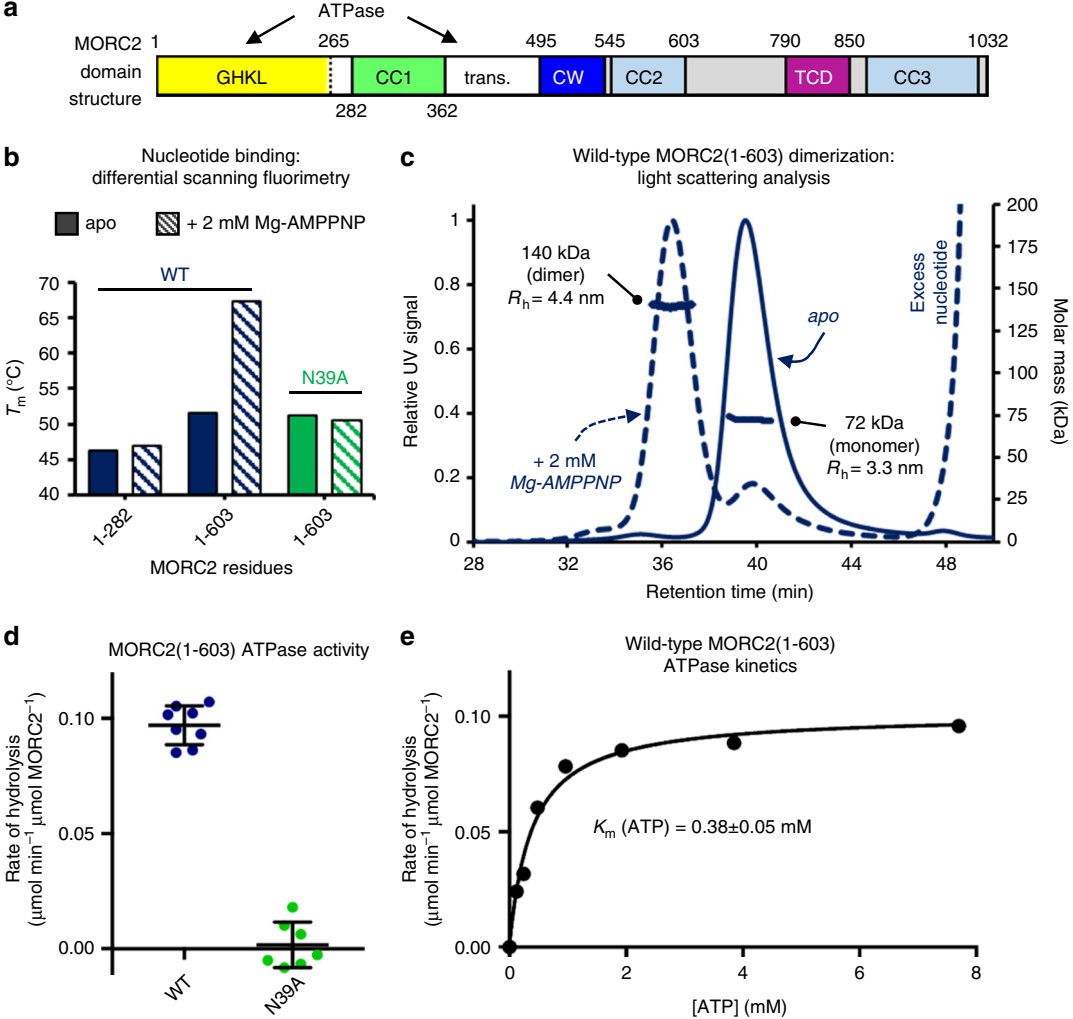

**Fig. 1** MORC2 is a GHKL-type ATPase. **a** Domain organization of human MORC2. The GHKL ATP binding domain and the transducer-like domain (trans.) together form the ATPase module, as marked. CC indicates a predicted coiled coil; CW indicates a CW-type zinc finger domain; TCD indicates a predicted tudor-chromodomain. **b** Fitted $T_m$s derived from differential scanning fluorimetry (DSF) for several MORC2 variants at 5 μM, in the absence (solid bar) and presence (striped bar) of 2 mM Mg-AMPPNP. This non-hydrolysable ATP analog significantly increases the thermal stability of wild-type (WT) MORC2 (1–603), while the N39A point mutant abrogates binding and WT MORC2(1–282) is stabilized to a much smaller extent than WT MORC2(1–603). Quoted $T_m$ values are an average of at least two replicates; note that the deviation between these measurements was <0.2 °C in all cases. **c** Portions of overlaid SEC-MALS UV traces for 40 μM WT MORC2(1–603) in the absence (solid line) and presence (dashed line) of 2 mM Mg-AMPPNP. The MALS data across the center of the major peaks in each case are shown on the right-hand axis, and are consistent with monomeric (expected mass: 70 kDa) and dimeric (expected mass: 140 kDa) species for the apo and AMPPNP-bound protein, respectively. Also shown are the fitted hydrodynamic radii obtained from QELS analysis of the peaks. The peak at 48 min in the AMPPNP-treated trace is the elution of excess (unbound) nucleotide. **d** Rate of ATP hydrolysis by wild-type (WT) and N39A MORC2(1–603) variants at 37 °C in the presence of 7.5 mM ATP, measured using an NADH-coupled continuous assay. Error bars represent standard deviation between measurements; $n = 8$ (WT), $n = 7$ (N39A). **e** Steady-state ATPase activity of 4 μM WT MORC2(1–603) at 37 °C fitted to a model of Michaelis–Menten kinetics

not stabilize the protein (Supplementary Fig. 1c). We then performed size-exclusion chromatography (SEC) coupled to both multi-angle light scattering (MALS) and quasi-elastic light scattering (QELS) to assess the oligomerization status. MALS analysis was consistent with the apo protein being monomeric, and dimerizing in the presence of 2 mM AMPPNP (Fig. 1c). QELS data showed that nucleotide-dependent dimerization was accompanied by an increase in the hydrodynamic radius ($R_h$) from 3.3 nm to 4.4 nm (Fig. 1c).

MORC2 was previously reported to have ATPase activity in an assay using cellular extracts and in which the D68A point mutant was used as a negative control[9]. We were unable to purify MORC2 constructs bearing the D68A mutation from either bacterial or eukaryotic cells, suggesting that it may cause

misfolding of the ATPase module. Since GHKL-type ATPases are usually inefficient enzymes, a robust negative control is essential to rule out background activity from more efficient, contaminating ATPases. Hence, we performed an ATPase assay with purified components, using the classical NADH-coupled system that has been used for DNA gyrase and Hsp90 in order to measure enzyme kinetics in continuous mode[26,27]. For the negative control, we mutated the highly conserved active site residue Asn39; the N39A mutation did not compromise the folding of MORC2, but abrogated binding of $Mg^{2+}$/AMPPNP according to DSF data and SEC (Fig. 1b and Supplementary Figs. 2 and 8). Purified wild-type MORC2(1–603) was found to have weak but measurable ATPase activity, while the N39A mutant was inactive (Fig. 1d). The kinetics we measured were

**Table 1 X-ray data collection and refinement statistics for crystal structures of MORC2(1–603) variants reported in this paper**

|  | WT | T424R | S87L |
|---|---|---|---|
| *Ligand* | AMP-PNP | AMP-PNP | ATP |
| *Data collection* |  |  |  |
| X-ray source | ESRF ID30b | ESRF ID29 | ESRF ID29 |
| Space group | $P12_1$ | $P12_1$ | $P12_1$ |
| Cell dimensions |  |  |  |
| $a, b, c$ (Å) | 66.17, 127.93, 80.19 | 69.61, 125.73, 81.53 | 69.85, 124.68, 80.36 |
| $\alpha, \beta, \gamma$ (°) | 90.0, 101.2, 90.0 | 90.0, 97.8, 90.0 | 90.0, 97.7, 90.0 |
| Wavelength (Å) | 0.972636 | 0.975999 | 0.975999 |
| Resolution (Å)[a] | 78.66–1.81 (1.84–1.81) | 80.76–2.57 (2.62–2.57) | 79.63–2.02 (2.05–2.02) |
| Observations | 789,315 | 165,185 | 309,681 |
| Unique reflections | 118,148 | 43,876 | 82,175 |
| $R_{merge}$ | 0.082 (0.901)[a] | 0.046 (0.599)[a] | 0.052 (0.648)[a] |
| $<I>/\sigma(I)$[a] | 13.6 (2.1) | 16.2 (2.1) | 13.5 (2.2) |
| Completeness (%)[a] | 98.9 (98.3) | 99.6 (99.4) | 91.9 (96.7) |
| Redundancy[a] | 6.9 (6.3) | 3.8 (3.7) | 3.8 (3.7) |
| *Refinement* |  |  |  |
| $R_{work}/R_{free}$ | 0.161/0.188 | 0.211/0.239 | 0.198/0.228 |
| No. of non-H atoms |  |  |  |
| Protein | 8708 | 8696 | 8713 |
| Ligand/ions | 66 | 66 | 67 |
| Solvent | 721 | 39 | 304 |
| Mean *B*-factors |  |  |  |
| Protein | 44.0 | 93.1 | 53.8 |
| Ligand/ions | 24.1 | 52.0 | 34.9 |
| Solvent | 45.4 | 57.2 | 47.0 |
| RMSD |  |  |  |
| Bond lengths (Å) | 0.012 | 0.014 | 0.015 |
| Bond angles (°) | 1.108 | 1.935 | 1.930 |
| Ramachandran stats |  |  |  |
| % Favored | 97.36 | 97.07 | 97.27 |
| % Allowed | 2.45 | 2.84 | 2.54 |
| % Outliers | 0.19 | 0.09 | 0.19 |
| *PDB code* | 5OF9 | 5OFA | 5OFB |

[a]Values in parentheses refer to the highest-resolution shell.

typical of GHKL ATPases, with a fitted $k_{cat}$ of 0.10 min$^{-1}$ and $K_m$ (ATP) of 378 ± 53 μM (Fig. 1e). Together, these data indicate that the wild-type MORC2 N-terminal ATPase module dimerizes upon ATP binding and that dimers dissociate upon ATP hydrolysis.

**Structure of the homodimeric N terminus of human MORC2.** Having isolated a MORC2 construct competent for nucleotide binding and hydrolysis, we sought to generate mechanistic insights into the biochemical properties of MORC2 and the molecular basis of MORC2-associated neuropathies via structural analysis. We obtained crystals of human MORC2(1–603) in the presence of a molar excess of AMPPNP. The structure was determined by molecular replacement, using the murine MORC3 ATPase module structure[11] as a search model. The asymmetric unit contained two MORC2 molecules and the structure was refined to 1.8 Å resolution (Table 1).

The overall architecture of the crystallized MORC2 fragment bound to AMPPNP is an almost symmetric, parallel homodimer resembling the letter M (Fig. 2a). Using the program HYDRO-PRO[28], we calculated the radius of gyration ($R_g$) of our model to

be 3.4 nm. This value is in good agreement with the hydrodynamic radius ($R_h$), 4.4 nm, obtained from QELS analysis of the AMPPNP-bound dimer in solution; theory states that for globular proteins, $R_g/R_h$ ~0.77[29]. A 2778 Å$^2$ surface from each monomer is buried at the dimer interface. Structural alignment of the ATPase modules of MORC2 and MORC3 gave an rmsd of 1.29 Å for 2200 backbone atoms, with 36% sequence identity.

The MORC2 ATPase module consists of two α-β-α sandwich domains, that we have distinguished as the GHKL domain (residues 1–265) and the transducer-like domain (residues 266–494, previously annotated as the S5 domain) due to its resemblance to the transducer domain of gyrase[30,31]. Notably, the β-sheet in the transducer-like domain contains an 80-amino acid antiparallel coiled-coil insertion, CC1 (residues 282–361), which forms a 6-nm projection emerging from the ATPase module. A similar insertion is predicted in MORC1, but is absent in other GHKL superfamily members. The transducer-like domain is capped by a helix-loop-helix motif that links to the CW domain (residues 495–545). This motif is disordered in the MORC3 structure and, moreover, the CW domain of MORC2 is in a completely different position and orientation relative to the ATPase module. Our MORC2 structures span residues 1–551, including all reported sites of neuropathy-causing mutations (Supplementary Fig. 3a,b). We did not observe electron density for the second predicted coiled coil, CC2 (residues 551–603). A tetrahedrally coordinated zinc atom carried over from the purification was observed bound to the CW domain. The presence of zinc in the MORC2 crystals was confirmed by X-ray fluorescence spectroscopy (Supplementary Fig. 3c).

MORC2 has a prototypical GHKL ATPase active site. One AMPPNP molecule, stabilized by an octahedrally coordinated Mg$^{2+}$ ion, is bound in the active site of both protomers. All critical residues involved in ATP binding and hydrolysis from the four signature motifs in the N-terminal GHKL ATP-binding domain[32] are conserved (Supplementary Fig. 3d,e): from Motif I (helix α2 in MORC2), Glu35 acts as a general base for water activation and Asn39 coordinates the Mg$^{2+}$ ion that templates the water-mediated interactions of the α-, β-, and γ- phosphates; from Motif II, Asp68 hydrogen bonds to the adenine-N6-amine and the bulky sidechain of Met73 stacks against the adenine ring, while Gly70 and Gly72 (the 'G1 box') appear to provide flexibility to the ensuing 'ATP lid'; from Motif III, Gly98, Gly101, and Gly103 form the 'G2 box' at the other end of the lid and Lys105 forms a salt bridge with the α-phosphate; and from Motif IV, Thr119 and Thr197 contribute to the stabilization of Motif II and the adenine ring, respectively. Lys427 from the transducer-like domain coordinates the γ-phosphate of AMPPNP, and forms a hydrogen bond with the same activated water nucleophile bound by Glu35. As in other GHKL family members, this conserved lysine from the transducer-like domain completes the functional ATPase.

**Nucleotide binding of MORC2 stabilizes dimer interface.** GHKL ATPases usually dimerize on binding ATP, but the composition and dynamics of the ATP lid that can close over the active site vary across the GHKL superfamily[32]. In the wild-type MORC2 structure, the ATP lid (residues 82–103) is in the closed conformation in both protomers, leaving only a narrow channel between the bound AMPPNP and the solvent. Aside from residues in the four motifs detailed above, protein–nucleotide interactions made by the sidechains of Ser87 (notably, a neuropathy mutation site) and Lys89 with the β-phosphate, and by the backbone atoms of Gln99 and Tyr100 with the γ-phosphate, stabilize the lid conformation (Fig. 2b). Residues in the lid form a

significant part of the dimer interface, with Ile82, Phe84, Arg90, Tyr100, and Asn102-forming hydrogen bonds and hydrophobic contacts with residues 12–24 of the other protomer. A loop in the transducer-like domain (residues 422–437) also contributes to the dimer interface. This loop coordinates the γ-phosphate of AMPPNP through Lys427 (and includes another neuropathy

mutation site, Thr424) (Fig. 2c). Residues 1–11 form the remaining contacts of the dimer interface, extending across all three layers of the GHKL domain of the other protomer. The majority of the dimer contacts are formed by loops that directly coordinate ATP and are likely to have a different, more flexible structure in the absence of ATP. This demonstrates how

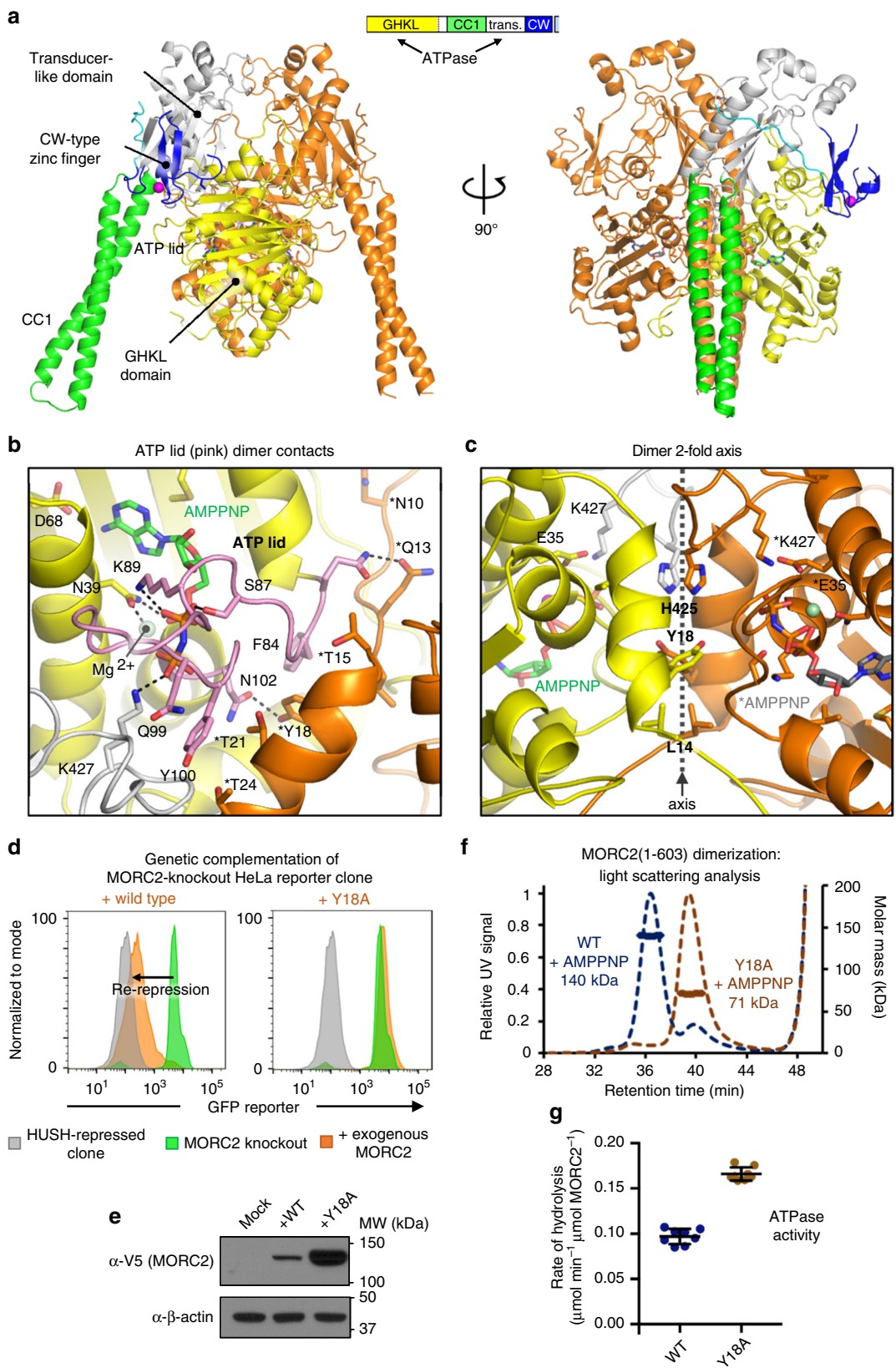

ATP binding and dimerization of MORC2 are coupled structurally.

**Dimerization of MORC2 is required for HUSH silencing**. The MORC2(1–603) N39A mutant is monomeric in solution and does not bind or hydrolyze ATP (Fig. 1b,d and Supplementary Fig. 1c, 2). Since ATP binding by the MORC2 ATPase module is coupled to dimerization, we conclude that the catalytically inactive N39A mutant does not form dimers via the ATPase module. We previously established a genetic complementation assay to assess the capacity of different disease-associated variants of MORC2 to rescue HUSH-dependent transgene silencing in MORC2 knockout (KO) cells. Briefly, we isolated clonal HeLa reporter cell lines bearing a HUSH-repressed GFP reporter. CRISPR-mediated KO of MORC2 in these clones led to the cells becoming GFP bright, allowing complementation with exogenous MORC2 variants, which can be monitored as GFP re-repression using FACS[4]. The lentiviral vector used expresses mCherry from an internal ribosome entry site (IRES), enabling us to control for multiplicity of infection (MOI) by monitoring mCherry. Using this assay, we previously found that the N39A mutant failed to rescue HUSH-dependent silencing[4]. Together with our biochemical data, this shows that ATP binding or dimerization of MORC2 (or both) is required for HUSH function.

To decouple the functional roles of ATP binding and dimerization, we used our MORC2 structure to design a mutation aimed at weakening the dimer interface without interfering with the ATP-binding site. The sidechain of Tyr18 makes extensive dimer contacts at the two-fold symmetry axis, but is not located in the ATP-binding pocket (Fig. 2c). Using the genetic complementation assay described above, we found that although the addition of exogenous V5-tagged wild-type MORC2 rescued HUSH silencing in MORC2-KO cells, the Y18A MORC2 variant failed to do so (Fig. 2d). Interestingly, the inactive MORC2 Y18A variant was expressed at a higher level than wild type despite the same MOI being used (Fig. 2e).

We then purified MORC2(1–603) Y18A and analyzed its stability and biochemical activities. Consistent with our design, the mutant was monomeric even in the presence of 2 mM AMPPNP according to SEC-MALS data (Fig. 2f). Despite its inability to form dimers, MORC2(1–603) Y18A was able to bind and hydrolyze ATP, with slightly elevated activity over the wild-type construct (Fig. 2g). This demonstrates that dimerization of the MORC2 N terminus is not required for ATP hydrolysis. Taken together, we conclude that ATP-dependent dimerization of the MORC2 ATPase module transduces HUSH silencing, and that ATP binding and hydrolysis are not sufficient.

**CC1 domain of MORC2 has rotational flexibility**. A striking feature of the MORC2 structure is the projection made by CC1

(residues 282–361) that emerges from the core ATPase module. The only other GHKL ATPase with a similar coiled-coil insertion predicted from its amino acid sequence is MORC1, for which no structure is available. Elevated *B*-factors in CC1 suggest local flexibility and the projections emerge at different angles in each protomer in the structure. The orientation of CC1 relative to the ATPase module also varies from crystal-to-crystal, leading to a variation of up to 19 Å in the position of the distal end of CC1 (Fig. 3a). Although the orientation of CC1 may be influenced by crystal contacts, a detailed examination of the structural variation reveals a cluster of hydrophobic residues (Phe284, Leu366, Phe368, Val416, Pro417, Leu419, Val420, Leu421, and Leu439) that may function as a 'greasy hinge' to enable rotational motion of CC1. Notably, this cluster is proximal to the dimer interface. Furthermore, Arg283 and Arg287, which flank the hydrophobic cluster at the base of CC1, form salt bridges across the dimer interface with Asp208 from the other protomer, and further along CC1, Lys356 interacts with Glu93 in the ATP lid (Fig. 3b). Based on these observations, we hypothesize that dimerization, and therefore ATP binding, may be coupled to the rotation of CC1, with the hydrophobic cluster at its base serving as a hinge.

**Distal end of CC1 contributes to MORC2 DNA-binding activity**. CC1 has a predominantly basic electrostatic surface, with 24 positively charged residues distributed across the surface of the coiled coil (Fig. 3c). MORC3 was shown to bind double-stranded DNA (dsDNA) through its ATPase module, but this interaction was autoinhibited by the CW domain[15]. Therefore, we sought to determine whether the MORC2 ATPase-CW cassette binds DNA, and whether the charged surface of CC1 contributes to DNA binding. We first performed electrophoretic mobility shift assays with nucleosome core particles (NCPs) and observed that wild-type MORC2(1–603) bound to both free DNA and nucleosomal DNA present in the NCP sample, with an apparent preference for free DNA (Fig. 3d).

Next, to assess the importance of CC1 in HUSH-dependent silencing, we examined the effect of a panel of charge reversal mutations in CC1 in the cell-based HUSH complementation assay. The charge reversal point mutations R319E, R344E, R351E, and R358E all rescued HUSH function in MORC2-KO cells, but R326E, R329E, and R333E (or combinations thereof) failed to do so (Fig. 3e and Supplementary Fig. 4a). Again, inactive variants were expressed at higher levels than active ones (Supplementary Fig. 4b). Residues 326, 329, and 333 form a positively charged patch near the distal end of the second α-helix of CC1. We therefore made a MORC2(1–603) triple mutant, R326E/R329E/ R333E, and compared its dsDNA binding to that of the WT construct. We confirmed that WT MORC2(1–603) bound to the canonical Widom 601 nucleosome positioning sequence with high apparent affinity, and observed a 'laddering' effect on the

**Fig. 2** ATP binding and dimerization of MORC2 are tightly coupled and required for HUSH-dependent transgene silencing. **a** Crystal structure of homodimeric human MORC2 residues 1–603 in complex with Mg-AMPPNP refined at 1.8 Å resolution. One protomer is colored according to the domain structure scheme (top), and the other is colored in orange. The protein is shown in cartoon representation, nucleotides are shown in stick representation, and metal ions are shown as spheres. Solvent molecules are not shown. **b, c** Nucleotide binding and dimerization are structurally coupled. Residues in the ATP lid (pink, residues 82–103), which covers the active site (**b**) and in a loop from the transducer-like domain (**c**) contribute to the interactions at the dimer interface. Key sidechains are shown in stick representation; labeled residues from the second protomer are marked with an asterisk. **d, e** Dimerization is critical for mediating HUSH-dependent transgene silencing activity. Expression of a MORC2 variant bearing an alanine substitution at a key residue in the dimer interface (Y18A) failed to rescue repression of a GFP reporter in MORC2 knockout cells, as assessed by FACS. Shown are the data from Day 12 post-transduction: the GFP reporter fluorescence of the HUSH-repressed clone is in gray; the MORC2 knockout is in green; the MORC2 knockout transduced with exogenous MORC2 variants is in orange (**d**). The lentiviral vector used expresses mCherry from an internal ribosome entry site (IRES), enabling control of viral titer by mCherry fluorescence measurement. Despite using the same MOI, the Y18A variant was expressed at higher levels than wild-type (WT) as assessed by a Western blot of cell lysates (**e**). **f, g** Y18A MORC2(1–603) does not undergo ATP-dependent dimerization, but is able to bind and hydrolyze ATP, based on SEC-MALS data in the presence of 2 mM Mg-AMPPNP (**f**) and ATPase assays (**g**). Error bars represent standard deviation between measurements; *n* = 8. The data for WT (blue) from Fig. 1c, d are shown for reference

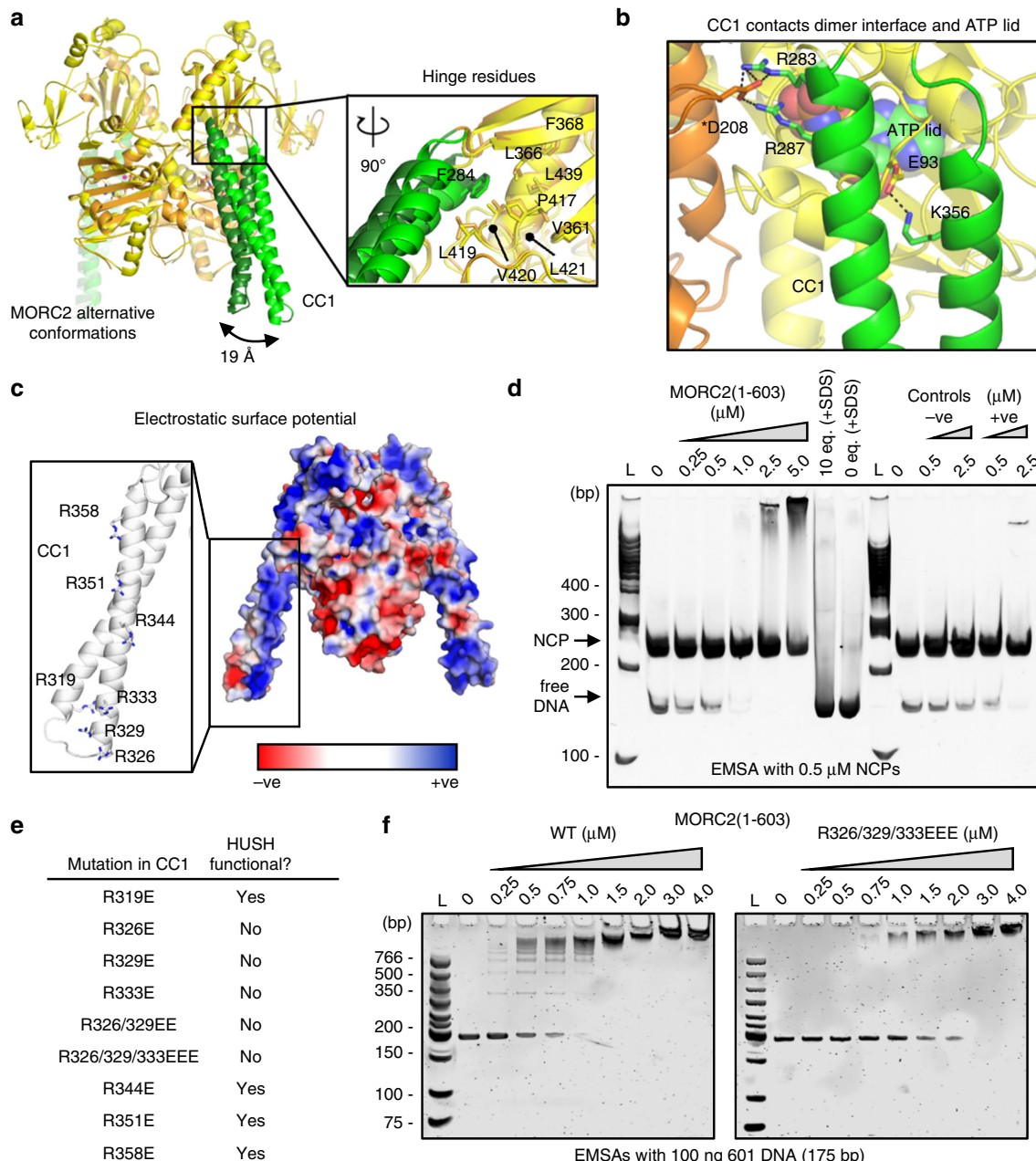

**Fig. 3** Novel coiled-coil insertion (CC1) in the GHKL ATPase module of MORC2 is hinged, highly charged, and important for DNA binding and HUSH function. **a** Superposition of MORC2(1–603) structures determined from two different crystals showing CC1 in different rotational states, leading to a difference of 19 Å in the position of the distal end of CC1. Inset: close-up of the boxed region, with the view rotated 90°, showing that hydrophobic residues (in stick representation) form a hinge at the base of CC1. **b** Close-up of the region boxed in panel A, highlighting in stick representation residues in CC1 that make polar interactions with the ATP lid and with the other MORC2 protomer, suggesting that the rotational motion of CC1 could be coupled to ATP binding and dimerization. **c** Electrostatic surface representation of the MORC2(1–603)-AMPPNP structure, made using the APBS PyMOL plug-in (red = negative charge, blue = positive charge). The positions of seven conserved arginines in CC1 are highlighted (inset). **d** MORC2(1–603) binds to dsDNA. EMSA with 500 nM nucleosome core particles (NCPs) in the presence of increasing concentrations of MORC2(1–603). Negative controls were performed with SDS-containing loading dye, which denatures proteins, and with a protein of similar size (−ve, concentrations at 0.5 and 2.5 μM). The positive control was HMGB1, a DNA-binding protein (+ve, concentrations at 0.5 and 2.5 μM). The gel was post-stained for DNA with SYBR Gold. L denotes the DNA ladder. **e** A charged patch at the tip of CC1 is required for HUSH-dependent silencing. Summary of FACS-based genetic complementation assays with charge reversal MORC2 mutants. See also Supplementary Figs. 4a,b. **f** CC1 contributes to the DNA-binding activity of MORC2(1–603). EMSAs with 100 ng 601 DNA in the presence of increasing concentrations of wild-type (WT, left) or triple mutant R326E/R329E/R333E (right) MORC2. The laddering effect on the DNA migration seen at low WT concentrations is absent in the case of the triple mutant, and the apparent affinity is weaker based on the slower disappearance of unbound DNA band with protein titration. Gels were post-stained for DNA with 2 μM SYTO62

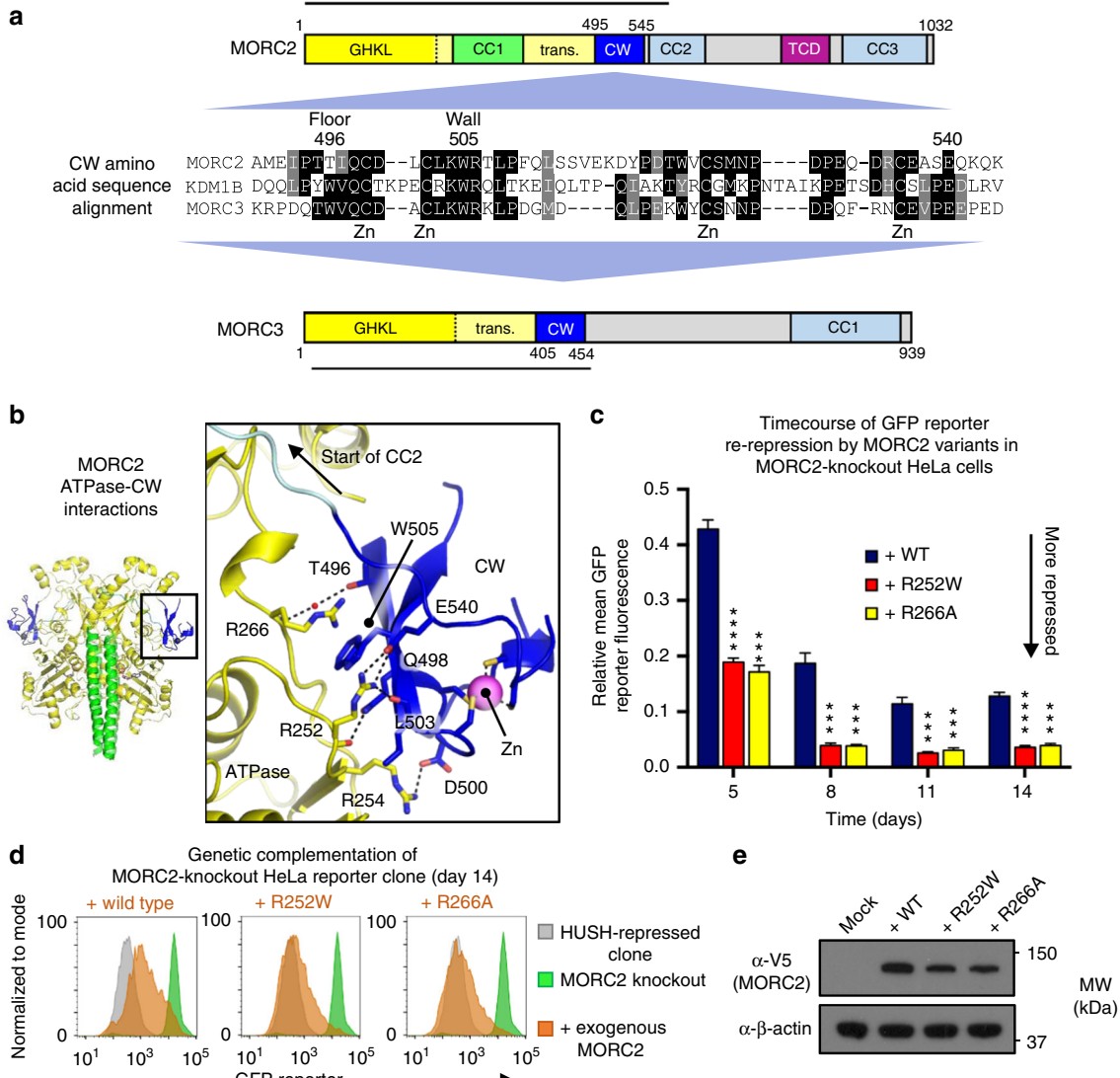

**Fig. 4** The degenerated histone-binding motif of the CW domain of MORC2 binds an Arg-rich surface on the ATPase module. **a** Amino acid sequence alignment of the CW domains of MORC2, lysine-specific demethylase 1B (KDM1B) and MORC3, shows that MORC2 lacks the 'floor' Trp residue important for H3K4me3 coordination by the other proteins. The sections of the MORC domain structure schemes marked with a line represent the regions resolved in crystal structures. 'Zn' marks the cysteine residues involved in zinc coordination. Residue numbers refer to human MORC2. **b** Details of the MORC2 ATPase–CW interaction. Key residues mentioned in the text are shown in stick representation and polar contacts (3.2 Å or less) are represented by dotted lines. **c–e** Weakening the ATPase–CW interaction in MORC2 hyperactivates HUSH-mediated transgene silencing. Time-course of the transgene re-repression by R252W and R266A MORC2 variants in the MORC2-knockout HeLa reporter clone (**c**). The bar chart shows the relative GFP fluorescence level (FACS-derived geometric mean) of complemented cells compared to the untreated cells (*$P < 0.05$; **$P < 0.01$; ***$P < 0.001$; ****$P < 0.0001$; ns, not significant, unpaired $t$-test compared to wild-type data, $n = 3$). Error bars represent the mean ± standard deviation from three biological replicates. Example FACS plots show that the R252W and R266A variants enhance the overall degree of GFP reporter repression relative to wild-type MORC2 (**d**). Shown are the data from Day 14 post-transduction: the GFP reporter fluorescence of the HUSH-repressed clone is in gray; the MORC2 knockout is in green; the MORC2 knockout transduced with exogenous MORC2 variants is in orange. Western blot validation of expression of the MORC2 variants (**e**)

DNA at the lowest (250–750 nM) protein concentrations. This is consistent with multiple DNA-binding surfaces on the protein, DNA modification and/or multiple proteins binding to a single piece of DNA. The triple CC1 charge reversal mutant still bound dsDNA, but with weaker apparent affinity, and no laddering of the DNA bands was observed (Fig. 3f). We confirmed that this mutant was folded and was able to bind and hydrolyze ATP (Supplementary Fig. 4c). The WT MORC2 GHKL domain alone (residues 1–282) also bound dsDNA, albeit with a much lower affinity and with no laddering, whereas the CW domain in isolation did not bind DNA in the EMSA (Supplementary Fig. 4d, e). Together, these data suggest that MORC2 binds dsDNA

through multiple sites including a positively charged surface near the distal end of the CC1 arm, and that the latter is required for transduction of HUSH-dependent silencing.

**CW domain of MORC2 regulates its HUSH effector function.** Several recent studies have shown that the CW domain of MORC3 binds H3K4me3 peptides selectively over histone 3 peptides with other epigenetic marks[11,14,15]. By contrast, the MORC2 CW domain does not bind to the H3K4me3 mark due to a missing tryptophan at the 'floor' of the CW aromatic cage (Thr496 in MORC2, Fig. 4a)[4,14]. Indeed, the MORC2 CW domain was found not to interact with any of the wide variety of

histone H3 and histone H4 peptides[14]. We confirmed that the lack of interaction with DNA and/or histones is not due to a folding defect or a reliance on the ATPase module for folding, since isolated $^{15}$N-labeled MORC2 CW domain gave well-dispersed peaks in a $^{1}$H, $^{15}$N-heteronuclear single quantum coherence experiment (Supplementary Fig. 5a).

The orientation of the CW domain relative to the ATPase module differs by approximately 180° in the MORC2 and MORC3 structures, with the degenerate histone-binding site of the MORC2 CW domain facing toward the ATPase module rather than toward solvent (Supplementary Fig. 5b). The CW domain binds an array of arginine residues in the transducer-like domain: conserved residue Trp505, providing the 'right wall' of the methyl-lysine-coordinating aromatic cage, forms a cation–π interaction with the sidechain of Arg266. Thr496 (the degenerated 'floor' residue) makes a water-mediated hydrogen bond with the backbone amide of Arg266. Asp500 forms a salt bridge with Arg254. Gln498 forms a hydrogen bond with the backbone carbonyl oxygen of Arg252. Glu540 forms a salt bridge with the Arg252 sidechain, which also forms a hydrogen bond with the backbone oxygen atom of Leu503 (Fig. 4b). The latter interactions are notable since a number of recent studies have shown that the R252W mutation causes CMT disease[16,17,20,21]. We recently demonstrated that this mutation causes hyperactivation of HUSH-dependent epigenetic silencing[4], leading to enhanced and accelerated re-repression of the GFP reporter in our functional assay. The R252W mutation, by removing the salt bridge to Glu540, may destabilize the ATPase–CW interface, which could account for the misregulation of MORC2 function in HUSH-dependent silencing. To test this hypothesis, we designed a mutation aimed at causing a similar structural defect, R266A, which disrupts the cation–π interaction with Trp505 described above. We performed a timecourse experiment, monitoring GFP reporter fluorescence in MORC2-KO cells after addition of the exogenous MORC2 variant. The R266A mutation recapitulated the hyper-repressive phenotype of R252W in the reporter clone tested (Fig. 4c, d). In contrast to inactive variants, these hyperactive variants were expressed at lower levels than wild type (Fig. 4e). These data support the notion that the ATPase–CW interaction in MORC2 has a regulatory function in HUSH transgene silencing.

In MORC3, the CW domain prevents binding of the ATPase module to DNA in the absence of the H3K4me3 peptide[15]. In MORC2, however, the CW domain does not inhibit DNA binding since MORC2(1–603) bound tightly to DNA despite the presence of an unliganded CW domain (Fig. 3d, f). We note that many of the sidechains forming key contacts in the ATPase–CW domain interfaces of MORC2 and MORC3 are not conserved in the two proteins. These non-conserved residues are Arg254, Arg266, and Thr496 in MORC2 and Glu184, Arg195, Lys216, Tyr217, Arg405, Arg444, and Asp454 in MORC3. Hence, it appears unlikely that the CW domain can bind to the MORC2 ATPase module in the same configuration as in MORC3, and vice versa. Together, our data show that the CW domain of MORC2 has a degenerate aromatic cage that explains its lack of binding to epigenetic marks on histone tails, and suggest that the association of the CW domain to the ATPase module antagonizes HUSH-dependent epigenetic silencing. Moreover, we conclude that MORC2 and MORC3 have evolved CW domains with distinct regulatory mechanisms.

**Disease mutations modulate the activities of MORC2**. We next tested whether MORC2 mutations reported to cause neuropathies affected the ATPase activity of MORC2. We purified MORC2 (1–603) variants containing the R252W, T424R, and S87L point mutations. All of the variants were folded and were thermally stabilized by addition of 2 mM $Mg^{2+}$/AMPPNP (Supplementary Figs. 2, 6a). We found a range of effects on ATPase activity (Fig. 5a). MORC2(1–603) bearing CMT mutation R252W[16,17,20,21] showed a small decrease in the rate of ATP hydrolysis. In contrast, SMA mutation T424R[19,22] increased ATPase activity by approximately three-fold. The S87L variant (for which the clinical diagnosis was CMT with SMA-like features[16,21]) eluted from a size-exclusion column as two species: a major species that eluted earlier than other variants and displayed elevated 260 nm absorbance (Supplementary Fig. 2), suggestive of dimerization and the presence of bound nucleotide(s), and a minor, presumably monomeric, species. This variant displayed low ATPase activity, near the detection threshold.

The R252W MORC2 variant hyperactivates HUSH-mediated transgene silencing[4], but has reduced ATPase activity in vitro. We used the timecourse HUSH functional assay in two distinct MORC2-KO GFP reporter clones (i.e., two different HUSH-repressed loci) to investigate further the correlation of these activities (Fig. 5b). S87L (which has reduced ATPase activity in vitro) also matched or outperformed wild-type MORC2 at each time point measured. Conversely, T424R (which has increased ATPase activity in vitro) was significantly less efficient at GFP reporter repression than wild-type at both loci (Fig. 5b and Supplementary Fig. 6b,c). Using SEC-MALS to investigate the oligomerization of S87L and T424R mutants, we confirmed that S87L forms constitutive N-terminal dimers with no exogenous addition of nucleotide, while T424R forms a mixture of monomers and dimers in the presence of 2 mM AMPPNP (Fig. 5c). Together, these data indicate that unlike the point mutants incompetent for ATP binding (N39A) or dimerization (Y18A), which altogether fail to transduce HUSH silencing, the disease-associated variants are all capable of ATP binding, dimerization, and hydrolysis. Further, we find that the efficiency of HUSH-dependent epigenetic silencing decreases as the rate of ATP hydrolysis increases. A summary of the properties of neuropathic and engineered MORC2 variants is shown in Table 2.

**Neuropathic mutations perturb MORC2 dimer interface**. Two MORC2 mutations, S87L and T424R, have been reported to cause congenital or infantile onset of neuropathies, distinct from the later onset that was reported for patients bearing the R252W (or other) mutations. The consequences of S87L and T424R mutations on the biochemical activities of MORC2 are drastic. The locations of these mutation sites—Ser87 in the ATP lid and Thr424 at the dimer interface—are also at functionally important regions in the structure and we determined the crystal structures of these variants to understand better the observed activities (Table 1). T424R MORC2 was co-crystallized with AMPPNP using the same protocol as for wild-type MORC2, but since S87L was dimeric and nucleotide-bound upon purification from insect cells, we determined its structure bound to ATP. The overall homodimeric structure of the two MORC2 disease variants was very similar to that of the wild type (Supplementary Fig. 7). The orientation of CC1 relative to the ATPase module varied in each protomer within the same range as in wild type. The ATP molecules bound to S87L MORC2 were found in a nearly identical conformation to AMPPNP in the wild-type and T424R structures, confirming that AMPPNP is a reasonable mimic of the natural nucleotide substrate in this case.

Ser87 is in the lid that covers bound ATP. Its sidechain hydroxyl forms a hydrogen bond with the β-phosphate of AMPPNP in the wild-type structure. In the S87L mutant, we found that the lid is partially missing in one protomer and has a

completely different conformation in the other. In the latter protomer, the lid forms additional contacts across the dimer interface in the S87L mutant (Fig. 5d). Leu87 itself forms apolar contacts with Asp141 from the other protomer, but more importantly, Arg90 forms a tight salt bridge with Glu17 across protomers. In the wild-type structure the Arg90 and Glu17 sidechains are 4–5 Å apart, but do not form a salt bridge. Instead, Lys86 can form a salt bridge with Asp141 from the other protomer in wild-type. The increased number of dimer contacts in the S87L mutant is reflected in an increased buried surface area at the dimer interface (3016 Å$^2$ buried per protomer versus 2778 Å$^2$ in wild-type). These observations provide a plausible structural basis for the observation that S87L forms more stable

ATP-bound dimers than wild-type, which in turn affects its cellular function.

The effect of T424R on the crystal structure of MORC2 is more subtle. The backbone structures of wild-type and T424R are essentially identical, including in the loop that contains the mutation (Fig. 5e). The arginine sidechain in the mutant does make an additional salt bridge across the dimer interface, with Glu27 from the other protomer. This additional contact may contribute to the dimer interface, but we did not observe any dimerization of T424R MORC2 during purification, suggesting that the mechanism of misregulating MORC2 is distinct from S87L. Moreover, the buried surface area at the dimer interface is actually decreased upon the T424R mutation (2527 Å$^2$ buried per

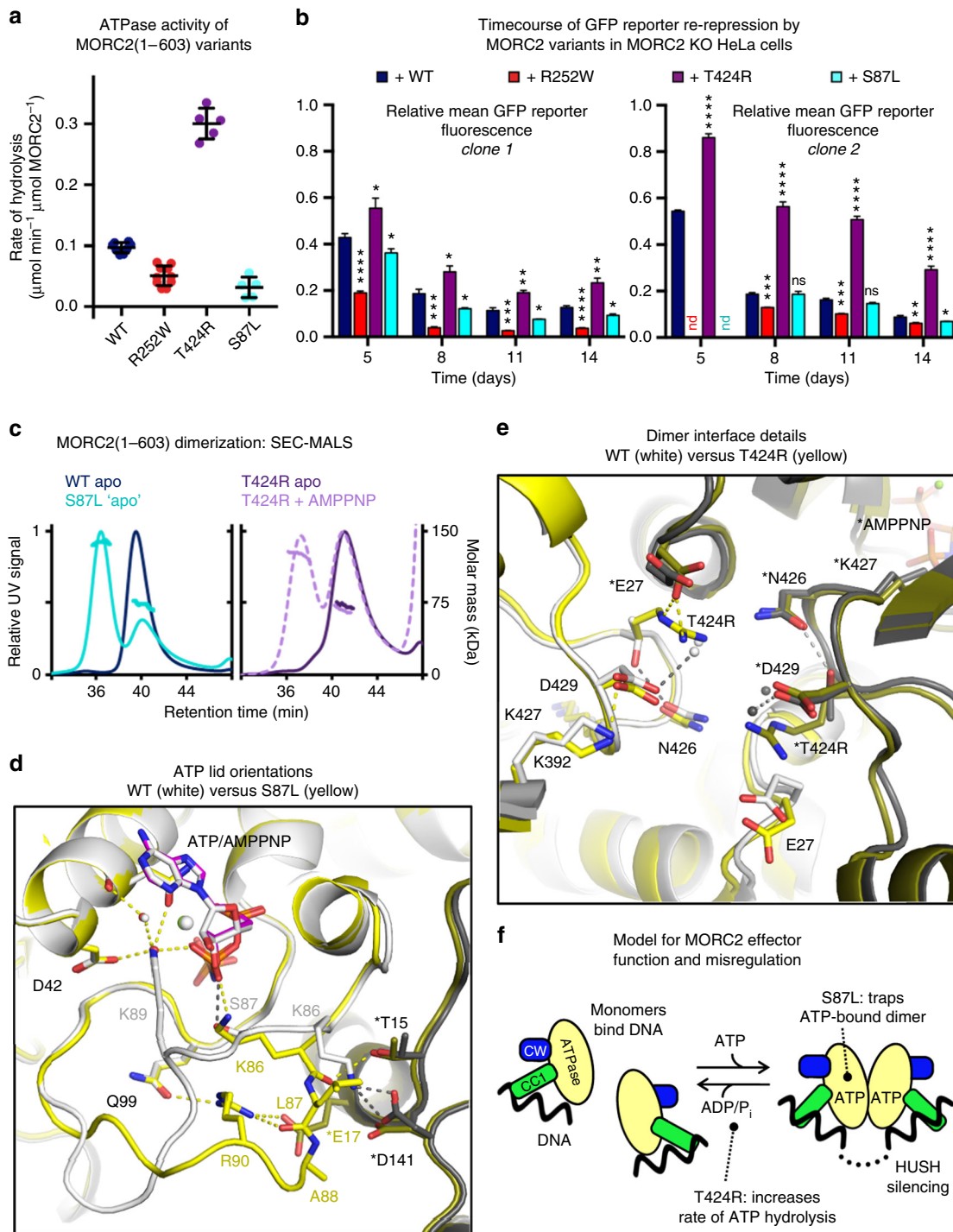

protomer versus 2778 Å$^2$ in wild type). We have described how ATP binding/hydrolysis is structurally coupled to dimerization/ dissociation. The contribution of the mutant Arg424 sidechain to the dimer interface, and its position just three residues away from a key active site residue Lys427, can be expected to alter the ATP-dependent dimerization dynamics of MORC2. Indeed, we found that the T424R variant forms a mixture of monomers and dimers in the presence of AMPPNP, and shows an elevated rate of ATP hydrolysis. This suggests that T424R dimers may form and dissociate more rapidly than in the wild type. It should be noted, however, that MORC2-associated neuropathies are subject to autosomal dominant inheritance. Therefore, our structures represent the physiologically less common species in which not one but both protomers bear the mutation. It may be that the effect on molecular function is subtly different in heterozygous MORC2 dimers. Together, these data show that S87L causes kinetic stabilization of MORC2 dimers, whereas T424R increases the rate of dimer assembly and disassembly (summarized in Fig. 5f). These two disease mechanisms are distinct from that of R252W, which we propose above to weaken the regulatory ATPase–CW interaction.

## Discussion

Genetic studies have established that MORC family proteins have fundamentally important functions in epigenetic silencing across eukaryotic species[1,4,5,8]. We recently identified MORC2 as an effector of the HUSH complex and showed that MORC2 contributes to chromatin compaction across HUSH target loci. The activity of MORC2 was dependent on ATP binding by its GHKL-type ATPase module[4]. Here, our structural and biochemical analyses provide evidence for how ATP binding and dimerization of MORC2 are coupled to each other.

To understand how the biochemical activity of MORC2 is related to its cellular function, a comparison to prototypical GHKL ATPases is informative. The $K_m$ for ATP and $k_{cat}$ of the MORC2 N-terminal fragment, 0.37 mM and 0.1 min$^{-1}$, respectively, are of comparable magnitude to those measured for recombinant constructs of *E. coli* DNA gyrase B (GyrB) (0.45 mM and 0.1 min$^{-1}$)[33], human Hsp90 (0.84 mM and 0.007 min$^{-1}$)[34], and MutL (0.09 mM and 0.4 min$^{-1}$)[35]. The $K_m$ of MORC3 has not been reported, but its activity at 3 mM ATP was 0.4–0.5 min$^{-1}$.[15] Hence, MORC2 and MORC3 resemble prototypical GHKL ATPases in that they bind ATP with relatively low affinity and hydrolyze ATP relatively slowly. Due to their low enzymatic turnover, GHKL ATPases are not known to function as motors or deliver a power stroke. Instead, ATP binding and hydrolysis function as conformational switches triggering dimer formation and dissociation, respectively[36]. Since MORC2 has similar enzymatic properties to other GHKL ATPases, we propose that its HUSH effector function arises from dimerization of its ATPase module triggered by ATP binding. Consistent with this model, a mutation outside the active site that inhibits dimerization of the ATPase module without affecting ATP binding/hydrolysis (Y18A) causes a loss of HUSH silencing (Fig. 2d–g). Since ATP hydrolysis leads to dissociation of the dimer, one corollary is that increasing the rate of hydrolysis should cause less efficient HUSH function, and decreasing the rate of hydrolysis should hyper-activate HUSH function (so long as the protein can undergo ATP-dependent dimerization). Indeed, this is what we found through analysis of the three neuropathy-associated MORC2 mutants: T424R, which hydrolyzes ATP more rapidly than wild-type, leads to less efficient HUSH silencing, whereas S87L and R252W, which hydrolyze ATP more slowly than wild-type, match or hyperactivate wild-type HUSH silencing (Fig. 5). We conclude that the lifetime of wild-type MORC2 ATPase dimers is tuned to enable cellular function. Indeed, the properties of engineered and naturally occurring MORC2 variants studied so far (summarized in Table 2) suggest that slight changes in the rate of ATP hydrolysis or dimer stability misregulate MORC2 cellular function by perturbing its N-terminal dimerization dynamics.

Our data suggest that the biological function of MORC2 arises from cycling between monomeric and dimeric conformational states at a finely tuned rate defined by ATP binding and hydrolysis—but what is the effect of MORC2 dimerization in the cell? The MORC2 N-terminal fragment binds dsDNA with high affinity as a monomer. Dimerization of two DNA-bound MORC2 molecules would bring two DNA duplexes together, which could promote DNA loop formation and chromatin compaction. MORC3 and SMCHD1 also bind DNA and have both been proposed to function as molecular clamps on DNA[11,37,38]. If MORC2 acts as a DNA clamp, the lifetimes of open and closed forms will be determined by the dimerization dynamics of the ATPase module. Structural and biochemical studies on full-length MORC2 are necessary to investigate whether MORC2 functions as a clamp and to better understand how it interacts with chromatin.

The structure of MORC2 contains a coiled-coil insertion in the ATPase module, CC1, which forms a positively charged 6-nm projection. A similar insertion is predicted in MORC1 (and in certain MORCs from other species including MORC1 from *C. elegans*, the sole annotated MORC in that species[3]), but not in other GHKL ATPases. MORC2 CC1 contributes to DNA binding, and charge reversal mutations at the distal end of CC1 cause a change in DNA-binding properties and loss of HUSH function. Comparison of MORC2 structures from different crystals shows that a cluster of hydrophobic residues, where CC1 emerges from

**Fig. 5** Neuropathy-associated mutations modulate the ATPase and HUSH-dependent silencing activities of MORC2 by perturbing its N-terminal dimerization dynamics. **a** Rate of ATP hydrolysis by wild-type (WT) and neuropathic variants of MORC2(1–603) at 37 °C and 7.5 mM ATP, measured using an NADH-coupled continuous assay. Error bars represent standard deviation between measurements; $n = 8$ (WT), $n = 10$ (R252W), $n = 5$ (T424R and S87L). The WT data are shown for reference and are the same as in Fig. 1d. **b** Assessing the efficiency of HUSH-mediated silencing by neuropathic MORC2 variants. Time-course of transgene re-repression by MORC2 variants in two distinct MORC2-knockout GFP reporter clones (i.e., two different HUSH-repressed loci). The bar chart shows the relative GFP fluorescence (FACS-derived geometric mean) of complemented cells compared to untreated cells (*$P < 0.05$, **$P < 0.01$, ***$P < 0.001$, ****$P < 0.0001$, ns: not significant, unpaired $t$-test compared to wild-type data). Error bars represent the mean ± standard deviation from three biological replicates. nd, not determined. See also Supplementary Figs. 6b,c. **c** Portions of overlaid SEC-MALS UV traces for neuropathic variants of MORC2(1–603). Left, 7.5 μM S87L MORC2(1–603) (light blue) as purified from insect cells with no additional nucleotide (WT apo data shown for comparison). Right, 7.5 μM T424R MORC2(1–603) in the absence (dark purple, solid line) and presence (light purple, dashed line) of 2 mM Mg-AMPPNP. **d,e** Neuropathic mutations S87L and T424R alter the modes of ATP binding and dimerization of the ATPase module. **d** Structure of the ATP lid (residues 82–103) of the S87L mutant. The structures of WT and S87L MORC2 dimers, bound to ATP and AMPPNP, respectively, were superimposed using the ATPase module of one protomer as the alignment reference. Residues in the second protomer are labeled with asterisks; colored olive and gray for S87L and WT, respectively. **e** Dimer interface near the site of SMA mutation T424R. The WT and T424R dimers were superimposed using both ATPase modules of each dimer as the alignment reference. The WT and mutant protomers are colored and labeled as in panel **c**. **f** Model of MORC2 effector function in HUSH silencing and its misregulation by neuropathic mutations S87L and T424R

**Table 2 Summary of the molecular consequences of neuropathic and structure-based mutants of MORC2**

| Mutation | Disease | Activity relative to wild-type MORC2 | | Position in structure | Proposed mechanism of MORC2 misregulation | References |
|---|---|---|---|---|---|---|
| | | **ATPase** | **HUSH** | | | |
| Y18A | — | Higher | No activity | Dimer interface | Does not dimerize | This paper |
| N39A | — | No activity | No activity | Active site | Cannot bind ATP | This paper;[4] |
| **S87L** | **CMT/SMA** | **Lower** | **Higher** | **ATP lid** | **Constitutive N-terminal dimerization** | [16,21] |
| R132L[a] | CMT | nd | nd | ATPase core | Destabilize ATPase[a] | [21] |
| E236G[a] | CMT | nd | nd | ATPase core | Destabilize ATPase[a] | [17] |
| **R252W** | **CMT** | **Lower** | **Higher** | **ATPase–CW interface** | **Destabilize ATPase-CW module** | [16,17,20,21,4] |
| R266A | — | nd | Higher | ATPase–CW interface | Destabilize ATPase-CW module | This paper |
| R333E[b] | — | nd | No activity | CC1 | DNA binding & CC1 functional defect | This paper |
| Q400R[a] | CMT | nd | nd | ATPase core | Destabilize ATPase[a] | [18] |
| **T424R** | **SMA** | **Higher** | **Lower** | **Dimer interface** | **Perturb dimerization dynamics** | [19,22] |
| D466N[a] | CMT | nd | nd | ATPase surface | Destabilize ATPase[a] | [18,23] |

Shown in bold are those disease mutants that we have investigated in this paper
nd, not determined
[a]These mutations have not been studied in this paper
[b]R333E is an example of several CC1 charge reversal mutants; see Fig. 3

the ATPase module near the dimer interface, provides rotational flexibility to CC1. Since residues in each CC1 form hydrogen bonds with the ATP lid and the other MORC2 protomer, ATP binding and dimerization of the ATPase module may be coupled to rotational motion of CC1, or to a change in the range of rotational states accessible to CC1. Additional mechanisms of coupling CC1 motions to ATP binding and dimerization may operate in the context of full-length MORC2. A structure of MORC2 in the absence of nucleotide is needed to shed further light on the motions of CC1 and the inter-domain dynamics of the system.

Despite similarities in the domain structure, several observations suggest that the mode of chromatin recruitment and molecular functions of different MORCs vary across the family. Firstly, MORC2 contains a tudor/chromo-like domain (TCD) not found in other MORCs. The MORC2 TCD is dispensable for effective HUSH function, suggesting that it is not required for recruitment of MORC2 to HUSH loci[4]. However, MORC2 may also be recruited to loci that are not HUSH targets[4], and the TCD may be involved in this. Secondly, the CW domains of MORC3 and MORC4 bind to H3K4me3, but the CW domains of MORC1 and MORC2 do not[14]. In the MORC2 structure, the degenerated histone-binding surface of the CW domain binds to an array of arginine residues in the ATPase module. Weakening these interactions hyperactivates HUSH silencing implying that the CW domain may have a regulatory function. Release of the CW domain may transmit a functionally important motion through the neighboring second coiled coil (CC2), the putative HUSH binding module[4], and could be triggered in WT MORC2 by another protein or unidentified epigenetic mark. Alternatively, the CW domain may function as a tandem reader (in concert with the TCD) by contributing additional affinity for a histone tail despite possessing very low affinity in isolation, as seen for example in CHD1[39]. Thirdly, sequence similarities between MORC2 and MORC1 suggest that the two proteins have structural and functional similarities. MORC1 has a similar domain structure to MORC2, including the CC1 insertion and a degenerated aromatic cage in the CW domain, lacking only the TCD. Human MORC1 is expressed exclusively in the male germline and the blastocyst, in which it silences active transposons at their H3K4me3-marked transcriptional start sites (TSSs)[5]. MORC2 is also found at TSSs in differentiated cells[4]. Notably, the CMT-

associated mutations R252W and D466N make the sequence of MORC2 more homologous to that of MORC1. Since the R252W MORC2 variant hyperactivates HUSH silencing, MORC1 may have greater repressive activity than MORC2. This could be advantageous because transposable elements are most actively expressed in the germline and during early development, due to the relief of epigenetic repression associated with the acquisition of pluripotency in these cells.

*MORC2* is the only gene in the MORC family in which mutations have been reported to cause neuropathies in humans. Here we have examined three disease mutations (S87L, R252W, and T424R) that cause neuropathies diagnosed as CMT and/or SMA. Our work provides insights on the basis of MORC2 misregulation in affected patients and identifies the link between the biochemical activities of MORC2 and its cellular function as an effector of HUSH. Modulation of putative HUSH-independent activities of MORC2 may also be important in determining disease outcome. For example, MORC2 has been associated with the activity of HDAC4[10], a protein known to be important in synaptic plasticity and as a transcriptional regulator in the central nervous system[40], and which was overexpressed in SMA-model mice and muscles of SMA patients[41]. More work will be needed to understand how *MORC2* mutations cause the range of clinical symptoms described, but it is interesting to note that the mutations causing pronounced changes in biochemical properties (S87L, which causes constitutive N-terminal dimerization, and T424R, which increases ATPase activity three-fold) are associated with congenital or infantile onset, unlike the R252W mutation, where the biochemical effect is more subtle and affected patients presented later.

Based on our observed relationships between its in vitro and in vivo activities, we conclude that MORC2 is part of a homeostatic system tuned such that a reduction in biochemical activity can cause a gain-of-function cellular phenotype, and vice versa. Similarly, mutations in *SMCHD1* cause BAMS, while others cause facioscapulohumeral muscular dystrophy type 2 (FSHD2)[24,25], with varied effects on in vitro ATPase activity[24]. These studies provide an interesting example in another GHKL ATPase of (i) pleiotropic disease outcomes resulting from mutations in the same gene, and (ii) gain of function cellular phenotypes resulting from decreased biochemical activity and vice versa. Since both gain and loss of MORC2 molecular function are linked to

neuropathies, there are, in theory, a large number of *MORC2* mutations that could cause disease including those that compromise the stability of the ATPase module. We would, therefore, predict that there are other *MORC2* mutations associated with undiagnosed neuropathies. Here we have described a molecular basis for understanding the MORC2 effector function in human epigenetic silencing and the varied mechanisms of misregulation that underlie MORC2-associated neuropathies.

## Methods

**Cell culture**. Sf9 insect cells (Expression Systems) were grown in Insect-XPRESS media (Lonza) with no antibiotics at 27 °C. HEK293T and HeLa cells (ECACC) were grown in IMDM plus 10% FCS, GlutaMAX (1×) and penicillin/streptomycin (100 U mL$^{-1}$). The cell lines were routinely tested for mycoplasma contamination using the MycoAlert detection kit (Lonza).

**Protein expression and purification**. Human MORC2 (Uniprot Q9Y6X9–1; residues 1–1032 and 1–603) were cloned by Gibson assembly into a modified pOET transfer vector (Oxford Expression Technologies) for baculovirus-driven production of N-terminally 3C protease-cleavable tandem-StrepII-tagged protein (Supplementary Table 1). *S. frugiperda* (Sf9) cells were co-transfected with this plasmid and linearized baculovirus genomic DNA (AB Vectors). Virus stocks were amplified with three rounds of infection. For expression, Sf9 cells at a density of 2–2.5 × 10$^6$ cells per mL were infected with 2.5% (v/v) third-passage (P3) virus and incubated with shaking at 27 °C for 50–60 h. All subsequent steps were performed at 4 °C. The cells were harvested by centrifugation, resuspended in lysis buffer containing 50 mM Tris, pH 8.0, 150 mM NaCl, 2 mM MgCl$_2$, 1 mM DTT, 1:10,000 (v/v) benzonase solution (Sigma), 1× cOmplete EDTA-free protease inhibitors (Roche), and flash–frozen in liquid nitrogen before storage at −80 °C. Upon thawing, the cells were lysed by sonication on ice. The ionic strength of the buffer was then increased by adding NaCl from a 5 M stock to a final concentration of 0.5 M. The lysate was clarified by centrifugation (1 h, 40,000× *g*). For a typical preparation at a ~ 200-μg scale, the protein-containing supernatant was bound to 500 μL pre-equilibrated StrepTactin Sepharose High Performance resin (GE Healthcare) for 1 h. The sample was then applied to a 2-ml Pierce Centrifuge Column (Thermo Scientific) and washed with at least 20 CV wash buffer (50 mM Tris, pH 8.0, 500 mM NaCl, 2 mM MgCl$_2$, and 1 mM DTT) before elution with 5 mM d-desthiobiotin in wash buffer. The buffer was exchanged to MORC2 gel filtration buffer (50 mM HEPES, pH 7.5, 150 mM NaCl, 2 mM MgCl$_2$, and 0.25 mM TCEP) with an Econo-Pac 10DG Desalting Column (Bio-Rad) and the tag cleaved by overnight incubation with 10 U PreScission Protease (GE Healthcare). Alternatively, proteins were eluted from the resin with PreScission cleavage (40 U). The GST-tagged protease was removed by incubating the cleaved sample twice with 100 μl glutathione agarose resin (Thermo Fisher Scientific). Final purification was achieved by SEC on a Superdex 200 increase (10/300) column (GE Healthcare). Variants were expressed and purified in the same way as for wild-type (WT) MORC2.

Human MORC2 (residues 1–282) was cloned into a pET15b vector for the production of N-terminally His-tagged protein and expressed in *E. coli* BL21(DE3) cells at 37 °C in 2xTY media containing 100 mg l$^{-1}$ ampicillin. Expression was induced at an OD$_{600}$ of 0.8 with 0.2 mM IPTG for 18 h at 18 °C. The culture was pelleted and resuspended in a buffer containing 50 mM Tris, pH 8.0, 500 mM NaCl, 10 mM imidazole, 1 mM DTT, 1:10,000 (v/v) benzonase solution (Sigma), and 1× cOmplete EDTA-free protease inhibitors (Roche), then flash–frozen in liquid nitrogen and stored at −80 °C. All subsequent steps were done at 4 °C. Further lysis was achieved by extensive sonication (3 × 3 min). The lysate was clarified by centrifugation and the protein-containing supernatant was bound to pre-equilibrated Ni-NTA beads (Generon) for 1 h. The beads were washed with at least 20 CV Ni wash buffer (50 mM Tris, pH 8.0, 500 mM NaCl, 10 mM imidazole, and 1 mM DTT) before a step-wise elution in batch mode with 3 × 5 CV of Ni wash buffer supplemented with 200 mM, 300 mM, and 500 mM imidazole. Further purification was performed with size-exclusion chromatography on a Superdex 200 increase (10/300) column (GE Healthcare) in MORC2 gel filtration buffer.

$^{15}$N-labeled human MORC2 CW domain (residues 490–546) was expressed as a His-SUMO-tagged fusion protein[4] in M9 media containing $^{15}$NH$_4$Cl (Sigma) as the sole nitrogen source. Purification was achieved by nickel affinity chromatography as described above, with the addition of 0.1 mM ZnSO$_4$ in all buffers, followed by tag cleavage with SENP-1 and SEC.

**Limited proteolysis**. Samples of 100 μl containing full-length MORC2 at ~0.1 mg ml$^{-1}$ in gel filtration buffer were incubated with 1% (w/w) trypsin (Pierce) at room temperature. Aliquots of 15 μl were removed at time intervals and the enzyme quenched by adding 5 μl of 4 × SDS-PAGE-loading buffer and heating the sample to 95 °C for 5 min. Samples of 10 μl were loaded on a NuPAGE 4–12% Bis-Tris gel (Thermo Fisher Scientific) in MES running buffer, separated at 200 V for 45 min, and then stained with Quick Coomassie stain (Generon).

**Differential scanning fluorimetry**. Samples of 10 μl containing MORC2 variants at 5 μM in the presence or absence of 2 mM AMPPNP or ADP/P$_i$ (Sigma) were prepared in gel filtration buffer, incubated on ice for 1 h, and then loaded into glass capillaries (Nanotemper) by capillary action. Intrinsic protein fluorescence at 330 and 350 nm was monitored between 15 and 90 °C in the Prometheus NT.48 instrument (Nanotemper), and the $T_m$ values calculated within the accompanying software by taking the turning point of the first derivative of the F$_{350}$:F$_{330}$ ratio as a function of temperature. Nucleotide concentrations were determined spectroscopically using $\varepsilon_{260}$ of 15.4 mM$^{-1}$ cm$^{-1}$.

**Light scattering**. Samples of 100 μL containing 0.5–3 mg ml$^{-1}$ MORC2(1–603) variants were analyzed by SEC at 293 K using a Superdex 200 (10/300) column (GE Healthcare) in gel-filtration buffer with a flow rate of 0.4 ml min$^{-1}$. Where indicated, 2 mM AMPPNP was added to the sample and incubated for 30 min on ice before injection. The SEC system was coupled to both MALS and QELS modules (Wyatt Technology). Light scattering analysis was performed in the ASTRA software package, using band broadening parameters obtained from a BSA standard run on the same day under identical conditions. MALS data were used to fit the average molar mass across the peak of interest (quoted to the nearest kDa) and QELS data were used to fit the hydrodynamic radii ($R_h$) of the species (quoted to the nearest 0.1 nm).

**ATP hydrolysis assays**. Initial assessment of ATPase activity of MORC2 constructs was done with an ATPase/GTPase assay kit (Sigma) for detecting the release of inorganic phosphate. Briefly, MORC2 at a final concentration of 3 μM was mixed with assay buffer (Sigma) and ATP (Sigma) added to a final concentration of 1 mM in a flat-bottomed 96-well clear plate (Corning). After 90 min at room temperature, the reaction was quenched with malachite green reagent and after a further 10 min incubation, the absorbance at 620 nm was read with a Clariostar plate reader (BMG LABTECH). For determining quantitative ATPase hydrolysis rates in continuous mode, the NADH-coupled system[27] was used to detect the evolution of ADP in 384-well microplates (Corning) at 37 °C. Assays were set up in a volume of 30 μL containing 0.35 mM NADH (VWR), 3 mM phosphoenolpyruvate (VWR), 8 U lactate dehydrogenase (Roche), 2.5 U pyruvate kinase (Roche), 7.5 mM ATP (Sigma), and 1.6–4 μM MORC2 variants. The components (apart from MORC2) were dissolved in an assay buffer containing 100 mM Tris, pH 8.0, and 20 mM MgCl$_2$. Blanks containing the protein buffer were included to account for background NADH decomposition over the course of the experiment. Pathlength-corrected NADH absorbance values at 340 nm were measured in a Clariostar plate reader at 37 °C, at 1-min intervals over a period of 90 min. Linear regression was used to fit the rate of NADH consumption at steady state. Rates are quoted as the consumption of NADH (μmol) per minute per μmol of MORC2, using the extinction coefficient $\varepsilon_{340}$(NADH) of 6.22 mM$^{-1}$cm$^{-1}$. The rates were blank-corrected and measured in triplicate. Non-linear regression analysis of $K_m$ and $V_{max}$ was done in GraphPad Prism; values are quoted ± standard error. ATP concentrations were determined spectroscopically using $\varepsilon_{260}$ of 15.4 mM$^{-1}$ cm$^{-1}$.

**X-ray crystallography**. WT MORC2(1–603) was concentrated to 5 mg ml$^{-1}$ (~70 μM) and incubated with ten molar equivalents AMPPNP in MORC2 gel filtration buffer. Crystals were grown at 18 °C by the sitting drop vapor diffusion method, by mixing the protein/nucleotide mixture at a 1:1 ratio with the reservoir solution optimized from the Morpheus screen[42]: 0.1 M bicine/Trizma, pH 8.5, 9% PEG4,000, 18% glycerol, and 0.12 M Morpheus alcohols mix (Molecular Dimensions) consisting of 0.02 M each of 1,6-hexanediol, 1-butanol, (RS)-1,2 propanediol, 2-propanol, 1,4-butanediol, and 1,3-propanediol. Pyramidal crystals appeared in 1–2 days and were frozen in liquid N$_2$ using 35% glycerol as a cryoprotectant. The procedure was the same for the S87L and T424R variants except that the precipitant concentrations for the best crystals were higher (10–11% PEG4000 and 20–22% glycerol). In the case of the S87L variant, the protein was at 4.5 mg ml$^{-1}$ and was concentrated from the earlier gel filtration peak (i.e., the nucleotide-bound dimeric fraction), which was assigned as ATP-bound; therefore, ATP was supplemented in place of AMPPNP. X-ray diffraction data were collected at 100 K at the European Synchrotron Radiation Facility (beamlines ID29 and ID30b) and processed using the autoPROC package[43]. Molecular replacement was done with Phaser[44] using the mouse MORC3 ATPase (PDB: 5IX2) as a search model[11] to place the ATPase module. The Zn atoms and coordinating residues from the CW domains were placed in real space, and the remainder built de novo in Coot[45]. Models were iteratively refined in Fourier space using REFMAC[46] or PHENIX[47]. The model building and real space refinement were done in Coot. Crystallographic data are summarized in Table 1. Structure figures were generated with PyMOL. The radius of gyration of the wild-type structure was calculated in the HYDRO-PRO software[28]. Portions of the final refined electron density maps are shown in Supplementary Fig. 9.

**Electrophoretic mobility shift assays**. Samples of 10 μl containing 100 ng of 601 mononucleosome positioning sequence DNA (a kind gift from T. Bartke, Helmholtz Zentrum, Munich) or 500 nM NCP (a kind gift from S. Tan, Penn State University) were incubated on ice with increasing concentrations of MORC2 variants in gel filtration buffer for 1 h. The reaction mixtures were loaded on a 6%

polyacrylamide gel that had been prerun on ice for 1 h at 150 V in 45 mM (0.5×) Tris-borate buffer. Electrophoresis was performed at 150 V on ice for 90 min. The gels were post-stained for DNA with 1× SYBR gold stain (Thermo Fisher Scientific), then visualized with the G-BOX system (Syngene), or with 2 μM SYTO 62 (Invitrogen), and then visualized with the Odyssey CLx system (LICOR).

**NMR spectroscopy**. A volume of 550 μl containing 60 μM $^{15}$N-labeled MORC2 CW domain in a buffer containing 50 mM HEPES, pH 7.5, 150 mM NaCl, 2 mM MgCl$_2$, 0.5 mM TCEP, 0.1 mM ZnSO$_4$, and 5% (v/v) D$_2$O was transferred to a 5-mm NMR tube (Norell). Spectra were recorded at 293 K on a Bruker 600 MHz Avance III spectrometer and processed in Topspin software.

**Flow cytometry**. The cells were analyzed on a FACSFortessa (BD Biosciences) instrument. Data were analyzed with FlowJo.

**Antibodies**. The following antibodies were used: mouse α-V5 (Abcam, ab27671, 1:5000 dilution), mouse α-β-actin (Sigma-Aldrich, A5316, 1:20,000 dilution), and donkey α-mouse HRP-conjugated antibody (Jackson ImmunoResearch, 715-035-150, 1:20,000 dilution). Validation of the antibodies is available online from the manufacturers.

**Immunoblotting**. The cells were lysed in 1% SDS plus 1:100 (v/v) benzonase solution (Sigma-Aldrich) for 15 min at room temperature, and then heated to 65 °C in SDS sample loading buffer for 5 min. Following separation by SDS-PAGE, the proteins were transferred to a PVDF membrane (Millipore), which was then blocked in 5% milk in PBS + 0.2% Tween-20. The membranes were probed overnight with the indicated primary antibodies, washed four times in PBS + 0.2% Tween-20, then incubated with HRP-conjugated secondary antibodies for 1 h at room temperature. Reactive bands were visualized using SuperSignal West Pico (Thermo Fisher Scientific). Uncropped western blots are shown in Supplementary Fig. 10.

**Lentiviral expression**. Exogenous gene expression in mammalian cells was performed using the vector pHRSIN-P$_{SFFV}$-V5-*MORC2*-IRES-mCherry-P$_{PGK}$-Hygro[4], with the wild-type V5-*MORC2* gene being replaced with the V5-*MORC2* variant of interest. The V5 tag included an SV40 nuclear localization signal with amino acid sequence PKKKRKV. Lentivirus particles were generated through the triple transfection of HEK293T cells with the lentiviral expression vector plus the two packaging plasmids, pCMVΔR8.91 and pMD.G, by using TransIT-293 transfection reagent (Mirus) according to the manufacturer's recommendations. Viral supernatant was typically harvested 48 h after transfection, cell debris was removed with a 0.45-μm filter, and target cells were transduced by spin infection at 800× *g* for 60 min. The transduced HeLa cells were selected with hygromycin at 100 μg ml$^{-1}$.

**MORC2 complementation in MORC2-knockout cells**. HeLa GFP MORC2 KO reporter clones were transduced in triplicate with an appropriate volume of either wild-type V5-*MORC2* or a variant V5-*MORC2* lentivirus, produced from the pHRSIN-P$_{SFFV}$-V5-*MORC2*-IRES-mCherry-P$_{PGK}$-Hygro construct. Viral supernatants were prepared, and cell transductions performed, as described above. Forty-eight hours post-transduction, the cells were divided into replicate plates, with one plate selected with 100 μg ml$^{-1}$ hygromycin, and the other not. To confirm that the same level of transduction was achieved across all wells, the unselected wells were FACS analyzed to determine the percentage of mCherry-positive cells on day 5 post-transduction. The selected wells were maintained as required, and were analyzed by FACS on the days indicated.

**Statistics**. No statistical methods were used to pre-determine the sample size, experiments were not randomized, and the investigators were not blinded to experimental outcomes. In all cases, ATPase activity data are represented as the mean ± s.d. of at least five replicates conducted across at least two independent experiments (i.e., different protein preparations). EMSA gels and DSF data shown are representative of at least two independent experiments. For the MORC2 complementation assays, data are representative of at least two independent experiments and statistical significance was assessed in GraphPad Prism.

**Data availability**. Crystal structures were deposited in the Protein Data Bank with IDs 5OF9, 5OFA, and 5OFB. Raw crystallographic data were deposited with SBGrid with data set numbers 457, 458, and 459. Other data are available from the corresponding authors upon reasonable request.

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

## Acknowledgements

Crystallographic data were collected on beamlines ID29 and ID30b at the European Synchrotron Radiation Facility (ESRF), Grenoble, France. We are grateful to Bart Van-Laer and David Flot at the ESRF for providing assistance in using the beamlines. We thank the scientists at the MRC Laboratory of Molecular Biology for support, particularly Chris Johnson, Minmin Yu, and Stefan Freund for their assistance with light scattering analysis, remote crystallographic data collection, and NMR spectroscopy, respectively. We also acknowledge Monique Merchant for help with insect cell culture and for collecting initial diffraction patterns, Daniil Prigozhin for the design of the insect cell expression vector backbone, and all members of the Modis lab for insightful discussions. This work was supported by a BBSRC Future Leader Fellowship to C.H.D. (BB/N011791/1), a Wellcome Trust Principal Research Fellowship to P.J.L. (101835/Z/13/Z), and a Wellcome Trust Senior Research Fellowship to Y.M. (101908/Z/13/Z).

## Author contributions

C.H.D., I.A.T., R.T.T., P.J.L., and Y.M. conceived the study. C.H.D., S.B., Y.L., M.S., and Y.M. performed the experiments. R.T.T. and I.A.T. contributed expression clones and cell lines. C.H.D, S.B., and Y.M. analyzed the data. C.H.D and Y.M. prepared the figures and wrote the manuscript with input from all authors. Y.M. supervised the study.

## Additional information

**Competing interests:** The authors declare no competing financial interests.

