## [Peer Review File · Nature Communications]

Reviewers' comments:

Reviewer #1 (Remarks to the Author):

Neuropathic mutations in MORC2 perturb GHKL ATPase dimerization dynamics and epigenetic silencing by multiple structural mechanisms.

Douse et al.

This paper describes characterisation of human mutations of MORC2 that cause neuropathies. As part of the study the authors have determined the crystal structures of WT MORC2 N-terminal domain (1-603) in complex with AMPPNP, T424R in complex with AMPPNP and S87L in complex with ATP. All of these structures are dimeric in the crystals, but the authors suggest that dimerisation is dynamic in solution, that the mutations perturb dimerization dynamics of the protein and that this is the mechanism through which they cause disease.

The authors have also studied other mutations such as N39A (which appears to no longer bind AMPPNP) and Y18A (which is at the dimerization interface and also appears to no longer bind AMPPNP).

Whilst this is overall a significant body of work and the conclusions are interesting, some more direct evidence for the dimerization state in solution would strengthen the manuscript considerably.

In particular the oligomerisation states need to be confirmed by a more direct method such as analytical ultracentrifugation or multi-angle light scattering.

The two-phase unfolding in the DSF thermal unfolding in the presence of AMPPNP (figure 1C) is not very convincing as a monomer/dimer equilibrium. The AMPNP could simply be stabilising a domain of the folded structure that then unfolds at a higher temperature.

In supplementary figure 5 the gel filtration profile of Y18A needs to be shown so as to demonstrate that the peak at 13.5ml is the monomeric form not the dimer that is seen in the crystals.

Other comments:

It is not possible to comment directly on the crystallography since there is no supplementary table 1 with the crystallographic structure determination statistics.

The dotted line which seems to represent a hydrogen bond in figure 2B between K89 and the AMPPNP appears to be at a strange angle. Is N39 interacting directly with the AMPPNP or through the Mg²⁺. It is not clear in the figure.

The EMSA experiment (fig 3D) has a negative control with SDS which would also be expected to denature the nucleosome and seems pointless as a negative control. This figure would benefit from the protein concentrations being shown in the same way as for figure 3F. For both of the experiments the length of the DNA is not stated.

The experiments in 4C, 4D and 5B are not described in the materials and methods. It is difficult to understand how you perform a 14 day time course in Hela cells.

It is not clear why the authors think that the T424R mutation perturbs dimerization dynamics since

the crystal structure is a static representation. The authors make the comment "can be expected to alter dimerization dynamics" but show no evidence for this.

Reviewer #2 (Remarks to the Author):

The manuscript reports the crystal structure and biochemical characterisation of the ~600 N-terminal amino acids of MORC2, which includes the ATPase domain. Structure-based mutants and disease associated mutations were investigated and the crystal structures of 2 mutant proteins (disease mutations) were also presented. The biochemical and structural features of these mutant proteins were importantly related to the capacity of MORC2 to epigenetically silence two different transgene arrays. The crystal structure of MORC2 has not been reported before and is novel, and will be of particular interest due to the mutations in this region of MORC2 that are found in Charcot-Marie-Tooth disease and Spinal muscular atrophy, which are studied here. The authors use their data to present an hypothesis for how MORC2 functions, and specifically how the ATP binding, dimerisation and subsequent monomerisation relates to silencing capacity. In my opinion this work is likely to influence thinking in the field, and be of interest to a wide range of readers from geneticists working on the relevant diseases, to structural biologists, biochemists and epigeneticists working on similar proteins.

There are a few areas where the work could be strengthened or clarified, as below.

Firstly, since dimerisation is an important, another method in addition to DSF should be used to analyse the critical mutant proteins e.g. analytical ultracentrifugation. The DSF should also be shown in full, meaning raw data as well as the tabulated data, even if the raw data goes into supplementary figures.

Secondly, all mutant proteins should be assessed for their dimerisation and ATPase activity. While some mutants such as in the coiled coil aren't expected to alter these features, for clarity they should be assessed and included to ensure the interpretation is appropriate.

It is noted that some mutants have a higher level of expression, despite the same MOI used to transduce cells, however no comment is made about the apparent drop in expression level when the mutation has the opposite effect on ATPase activity. Could the authors include a comment on this?

For easier visualisation, could the lid be shown in a different colour in the structure in Fig 2 as it is difficult to discern.

In most instances, the number of replicates aren't mentioned in the figure legends. Can the authors include the number of replicates, and also the statistical test used to obtain the p values provided.

Reviewer #3 (Remarks to the Author):

This manuscript presents a comprehensive structure-function analysis of MORC2, a protein involved in gene silencing and human neuropathies. Using differential scanning fluorimetry and ATPase assays the authors show that MORC2 protein forms a dimer that is stabilized by ATP and magnesium binding and that it has a relatively modest ATPase activity. The authors further present the structure of the N-terminal ATPase part of MORC2 including the ATPase with its transducer domain and a coiled-coil insertion as well as the following CW domain. The wild type and mutant structures presented are of high quality and provide a large

amount of information on MORC2 that the manuscript leverages to elucidate the mechanism of MORC2 in gene silencing. To this end the authors combine their biochemical assays with a HeLa reporter system that measures gene silencing. They find that dimerization is required for silencing and that the coiled-coil domain influences DNA binding activity of MORC2 and is involved in gene silencing. Furthermore they analyze several disease-related mutations of MORC2 and find interesting and diverse biochemical behavior, which suggests that MORC2 activity is fine-tuned to a specific activity and that loss or gain of biochemical activities disturb its biological function.

This manuscript provides a very valuable and detailed structure-function study on MORC2 and will be an important reference for research on MORC-related proteins and the field of mammalian gene silencing. It also provides a useful basis for developing strategies to tackle the diseases related to MORC2 mutations.

From my perspective as a structural biologist this manuscript presents a solid body of work that is well written and presented and absolutely deserves to be published.

Points to address:

1. The authors should provide a model in their last figure that summarized their results and illustrates how MORC2 activity and structure relates to gene silencing.
2. The yellow shades of color in the ATPase domain are impossible to see in the figures. It would be worth giving the transducer domain a more distinctive color.
3. The side chain statistics look a bit weak for the mutant structures. The authors should verify that there are no gross mistakes in the rotamer assignments.

Reviewer #1 (Remarks to the Author):

This paper describes characterisation of human mutations of MORC2 that cause neuropathies. As part of the study the authors have determined the crystal structures of WT MORC2 N-terminal domain (1-603) in complex with AMPPNP, T424R in complex with AMPPNP and S87L in complex with ATP. All of these structures are dimeric in the crystals, but the authors suggest that dimerisation is dynamic in solution, that the mutations perturb dimerization dynamics of the protein and that this is the mechanism through which they cause disease. The authors have also studied other mutations such as N39A (which appears to no longer bind AMPPNP) and Y18A (which is at the dimerization interface and also appears to no longer bind AMPPNP).

Whilst this is overall a significant body of work and the conclusions are interesting, some more direct evidence for the dimerization state in solution would strengthen the manuscript considerably.

We thank the reviewer for his/her constructive feedback and agree with the comment about dimerization; see below.

-In particular the oligomerisation states need to be confirmed by a more direct method such as analytical ultracentrifugation or multi-angle light scattering.

In response to this comment we performed size-exclusion chromatography (SEC) coupled to both multi-angle light scattering (MALS) and quasi-elastic light scattering (QELS) for the WT and dimerization mutant (Y18A), in the presence and absence of 2 mM AMPPNP. The results are presented in Figs. 1C and 2E in the revised manuscript.

The MALS data in Fig. 1C clearly show that the apo WT protein is monomeric in the absence of nucleotide, and dimeric in the presence of AMPPNP. Furthermore, using the QELS data we were able to fit the hydrodynamic radius of the solution-state dimeric species (R_h , 4.4 nm) which matches well the predicted hydrodynamic properties of the dimeric crystal structure: the calculated radius of gyration (R_g) from the WT pdb is 3.4 nm; theory states that for globular particles, $R_g/R_h \sim 0.8$.

In Fig. 2E, we show that the Y18A construct is strictly monomeric, even in the presence of 2 mM AMPPNP. The nano-differential scanning fluorimetry (DSF) data are still summarized in Fig. 1B (in revised and clearer histogram form), but the manuscript no longer relies on the DSF data as evidence of dimerization. The DSF curves have been moved to the supplementary data (Supplementary Figures 1C and 8).

-The two-phase unfolding in the DSF thermal unfolding in the presence of AMPPNP (figure 1C) is not very convincing as a monomer/dimer equilibrium. The AMPNP could simply be stabilising a domain of the folded structure that then unfolds at a higher temperature.

We agree on reflection that DSF is not the most elegant way of probing dimerization. We have removed fig 1C and replaced it with light scattering analysis, as described above. We have now restricted our interpretation of DSF data to being a readout of nucleotide binding, both in the figures and in the text throughout.

-In supplementary figure 5 the gel filtration profile of Y18A needs to be shown so as to demonstrate that the peak at 13.5ml is the monomeric form not the dimer that is seen in the crystals.

The SEC-MALS data on Y18A (Fig 2E) now clearly shows that the gel filtration peak corresponds to a monomer. Additionally, we have now added Supp. Fig. 2, which shows all

of the gel filtration profiles of the purified variants, including Y18A, for ease of reference and comparison.

-It is not possible to comment directly on the crystallography since there is no supplementary table 1 with the crystallographic structure determination statistics.

We regret that the reviewer was not able to access Supp. Table 1. The table was included in the original submission in a separate file to the main supplementary information file (in Word format), as required by the online submission system.

-The dotted line which seems to represent a hydrogen bond in figure 2B between K89 and the AMPPNP appears to be at a strange angle. Is N39 interacting directly with the AMPPNP or through the Mg²⁺. It is not clear in the figure.

We have checked the hydrogen bond in question and found the angle made by atoms K89(Cε), K89(Nε) and the beta phosphate oxygen of AMPPNP is 112°. The distance between the latter two atoms is 2.7 Å. These numbers are typical and close to ideal geometry. Regarding the interactions of N39, the amide carbonyl contributes to the octahedral coordination of the Mg²⁺ ion, while the amide nitrogen interacts directly with the alpha-phosphate of AMPPNP. To clarify the hydrogen bonding connectivity in the figure, we have added dashed lines between N39, the Mg²⁺ ion and the AMPPNP alpha-phosphate. The reviewer may have been confused by the fact that the alpha and beta phosphates are overlapping in this figure panel, but this was unavoidable in order to show the ATP lid in an optimal view, which is the aim of the figure. However, we have added Supp. Fig. 3d to show a more complete map of the many interactions with AMPPNP within the active site, in 2D schematic form.

-The EMSA experiment (fig 3D) has a negative control with SDS which would also be expected to denature the nucleosome and seems pointless as a negative control. This figure would benefit from the protein concentrations being shown in the same way as for figure 3F. For both of the experiments the length of the DNA is not stated.

We agree that SDS would denature the nucleosome. Since MORC2 binds to the free DNA present in the sample, we included this control to test whether free DNA binding was dependent on MORC2 being folded rather than some overall nonspecific charge effect. Since there is no electrophoretic mobility shift in this lane, we conclude that MORC2 must, as expected, be folded to cause the shift. Regarding the length of the DNA and the protein concentrations, we have now stated these in the revised manuscript.

-The experiments in 4C, 4D and 5B are not described in the materials and methods. It is difficult to understand how you perform a 14 day time course in HeLa cells.

We thank the reviewer for pointing out this omission and have added a section in the Materials and Methods describing the time course experiments. The HeLa cells were maintained with 100 µg/ml hygromycin selection and media changes as required during the time course.

-It is not clear why the authors think that the T424R mutation perturbs dimerization dynamics since the crystal structure is a static representation. The authors make the comment “can be expected to alter dimerization dynamics” but show no evidence for this.

To address this comment, in addition to the light scattering analysis we performed with the WT and Y18A variants, we have now added SEC-MALS of two key neuropathic mutations, S87L and T424R. Our new data, presented in Fig. 5C, show that MORC2-S87L forms

constitutive N-terminal dimers with no exogenous addition of nucleotide (Fig. 5C, left). MORC2-T424R in the presence of 2 mM AMPPNP forms a mixture of monomers and dimers (Fig. 5C, right). The new data illustrate that the T424R variant dimerizes in solution in a nucleotide-dependent manner. Since its ATP hydrolysis activity – which we propose leads to dimers to dissociate into monomers as in WT – is elevated (Fig. 5A), we conclude that the rate of interconversion between monomer and dimer is likely increased in the T424R variant with respect to the WT protein.

Reviewer #2 (Remarks to the Author):

The manuscript reports the crystal structure and biochemical characterisation of the ~600 N-terminal amino acids of MORC2, which includes the ATPase domain. Structure-based mutants and disease associated mutations were investigated and the crystal structures of 2 mutant proteins (disease mutations) were also presented. The biochemical and structural features of these mutant proteins were importantly related to the capacity of MORC2 to epigenetically silence two different transgene arrays. The crystal structure of MORC2 has not been reported before and is novel, and will be of particular interest due to the mutations in this region of MORC2 that are found in Charcot-Marie-Tooth disease and Spinal muscular atrophy, which are studied here. The authors use their data to present an hypothesis for how MORC2 functions, and specifically how the ATP binding, dimerisation and subsequent monomerisation relates to silencing capacity. In my opinion this work is likely to influence thinking in the field, and be of interest to a wide range of readers from geneticists working on the relevant diseases, to structural biologists, biochemists and epigeneticists working on similar proteins.

There are a few areas where the work could be strengthened or clarified, as below.

We thank the reviewer for his/her comments and feedback.

-Firstly, since dimerisation is an important, another method in addition to DSF should be used to analyse the critical mutant proteins e.g. analytical ultracentrifugation. The DSF should also be shown in full, meaning raw data as well as the tabulated data, even if the raw data goes into supplementary figures.

We agree on reflection that DSF is not the ideal method for analysing dimerization. As described above in the response to related comments from Reviewer 1, we have performed size exclusion chromatography (SEC) coupled to both multi-angle light scattering (MALS) and quasi-elastic light scattering (QELS) for the WT and dimerization mutant (Y18A), in the presence and absence of 2 mM AMPPNP (see Figs. 1C and 2E). We have also performed SEC-MALS with two of the key neuropathic mutants (S87L & T424R, see Fig. 5C). The SEC-MALS data provide a clear and unambiguous readout on the oligomeric state of the MORC2 variants, which turned out to be consistent with our previous DSF data (see also response to Reviewer 1).

Regarding the DSF data, we have now added all the raw data to the Supp. Figures (Supplementary Figures 1C and 8).

-Secondly, all mutant proteins should be assessed for their dimerisation and ATPase activity. While some mutants such as in the coiled coil aren't expected to alter these features, for clarity they should be assessed and included to ensure the interpretation is appropriate.

We have added new Supp. Figs. 2 & 8 which show all of the gel filtration profiles and DSF data for the purified variants. We have conducted ATPase and DSF assays with the coiled

coil mutant (R326E/R329E/R333E, see Supp. Fig. 4c), confirming that the protein stability, nucleotide binding and hydrolysis are only marginally affected by these mutations.

Altogether, this means that the manuscript now contains size-exclusion chromatography, ATP hydrolysis assays and DSF data for all of the MORC2 variants which we have studied, in addition to SEC-MALS data for several key site-directed and disease mutants, as requested.

-It is noted that some mutants have a higher level of expression, despite the same MOI used to transduce cells, however no comment is made about the apparent drop in expression level when the mutation has the opposite effect on ATPase activity. Could the authors include a comment on this?

We think the observation that hypoactive or non-functional variants are expressed at higher levels whereas hyperactive variants are expressed at lower levels could be interesting. This effect may be HUSH-dependent: i.e. it could be that a more active HUSH is silencing the lentivirus used to (exogenously) express MORC2 constructs, but it could also be that the more active variants are more toxic to cells and thus the cells that survive are those that have found a way of restricting expression of those variants. We have added a comment as requested stating that hyperactive variants were expressed at lower levels than wild-type and referring to Fig. 4E (p. 10).

-For easier visualisation, could the lid be shown in a different colour in the structure in Fig 2 as it is difficult to discern.

This has been changed to light pink.

-In most instances, the number of replicates aren't mentioned in the figure legends. Can the authors include the number of replicates, and also the statistical test used to obtain the p values provided.

This information has been added, where missing, throughout.

Reviewer #3 (Remarks to the Author):

This manuscript presents a comprehensive structure-function analysis of MORC2, a protein involved in gene silencing and human neuropathies. Using differential scanning fluorimetry and ATPase assays the authors show that MORC2 protein forms a dimer that is stabilized by ATP and magnesium binding and that it has a relatively modest ATPase activity. The authors further present the structure of the N-terminal ATPase part of MORC2 including the ATPase with its transducer domain and a coiled-coil insertion as well as the following CW domain. The wild type and mutant structures presented are of high quality and provide a large amount of information on MORC2 that the manuscript leverages to elucidate the mechanism of MORC2 in gene silencing. To this end the authors combine their biochemical assays with a HeLa reporter system that measures gene silencing.

They find that dimerization is required for silencing and that the coiled-coil domain influences DNA binding activity of MORC2 and is involved in gene silencing. Furthermore they analyze several disease-related mutations of MORC2 and find interesting and diverse biochemical behavior, which suggests that MORC2 activity is fine-tuned to a specific activity and that loss or gain of biochemical activities disturb its biological function.

This manuscript provides a very valuable and detailed structure-function study on MORC2 and will be an important reference for research on MORC-related proteins and the field of mammalian gene silencing. It also provides a useful basis for developing strategies to tackle the diseases related to MORC2 mutations.

From my perspective as a structural biologist this manuscript presents a solid body of work that is well written and presented and absolutely deserves to be published.

We thank the reviewer for these comments.

Points to address:

-1. The authors should provide a model in their last figure that summarized their results and illustrates how MORC2 activity and structure relates to gene silencing.

We have added such a model in a new panel, Fig. 5F.

-2. The yellow shades of color in the ATPase domain are impossible to see in the figures. It would be worth giving the transducer domain a more distinctive color.

We have changed the color of the transducer domain within the ATP module to white (light grey), so that different shades of yellow are no longer used.

-3. The side chain statistics look a bit weak for the mutant structures. The authors should verify that there are no gross mistakes in the rotamer assignments.

We reviewed the side chain rotamer outliers individually in the mutant structures. We can confirm there are no gross mistakes. In the majority of rotamer outliers, the density is of high quality and the outliers can be explained by polar or stacking interactions that pull the side chain out of its favoured rotamer. However, in a subset of side chains in each mutant structure, there was some ambiguity in the side chain conformation and we therefore adjusted the sidechain orientations. These adjustments reduced the fraction of rotamer outliers in the T424R structure (5OFA) from 9.6% to 6.7% (and from the 18th to the 36th percentile relative to structures of similar resolution), in the S87L structure (5OFB) from 6.9% to 5.7% (or from the 12th to the 18th percentile relative to structures of similar resolution). We have uploaded and submitted revised coordinates to the PDB for entries 5OFA and 5OFB and thank the reviewer for bringing this to our attention. Validation reports for the updated coordinates were included in this submission.